# Elevated H3K79 homocysteinylation causes abnormal gene expression during neural development and subsequent neural tube defects

Qin Zhang[1], Baoling Bai[1], Xinyu Mei[2], Chunlei Wan[1], Haiyan Cao[1], Dan Li[1,3], Shan Wang[1], Min Zhang ⬤ [1], Zhigang Wang ⬤ [4], Jianxin Wu[1], Hongyan Wang[2], Junsheng Huo[5], Gangqiang Ding[5], Jianyuan Zhao[6], Qiu Xie[4], Li Wang ⬤ [1], Zhiyong Qiu[1], Shiming Zhao[2] & Ting Zhang[1]

Neural tube defects (NTDs) are serious congenital malformations. Excessive maternal homocysteine (Hcy) increases the risk of NTDs, while its mechanism remains elusive. Here we report the role of histone homocysteinylation in neural tube closure (NTC). A total of 39 histone homocysteinylation sites are identified in samples from human embryonic brain tissue using mass spectrometry. Elevated levels of histone KHcy and H3K79Hcy are detected at increased cellular Hcy levels in human fetal brains. Using ChIP-seq and RNA-seq assays, we demonstrate that an increase in H3K79Hcy level down-regulates the expression of selected NTC-related genes including *Cecr2*, *Smarca4*, and *Dnmt3b*. In human NTDs brain tissues, decrease in expression of *CECR2, SMARCA4,* and *DNMT3B* is also detected along with high levels of Hcy and H3K79Hcy. Our results suggest that higher levels of Hcy contribute to the onset of NTDs through up-regulation of histone H3K79Hcy, leading to abnormal expressions of selected NTC-related genes.

[1] Beijing Municipal Key Laboratory of Child Development and Nutriomics, Capital Institute of Pediatrics, 100020 Beijing, China. [2] Obstetrics & Gynecology Hospital of Fudan University, State Key Lab of Genetic, Engineering and Institutes of Biomedical Sciences, 200433 Shanghai, China. [3] Weifang Medical University, 261053 Weifang, China. [4] Chinese Academy of Medical Sciences & Peking Union Medical College, 100005 Beijing, China. [5] Key Laboratory of Trace Element Nutrition of National Health and Family Planning Commission of the People's Republic of China, National Institute for Nutrition and Health, Chinese Center for Disease Control and Prevention, 102206 Beijing, China. [6] State Key Laboratory of Genetic Engineering and School of Life Sciences, Fudan University, 200438 Shanghai, China. These authors contributed equally: Qin Zhang, Baoling Bai, Xinyu Mei. Correspondence and requests for materials should be addressed to S.Z. (email: zhaosm@fudan.edu.cn) or to T.Z. (email: Zhangtingcv@126.com)

Human neural tube defects (NTDs) are common, severe, and costly birth defects that arise between the third and fourth weeks of embryogenesis due to partial or complete failure of neural tube closure (NTC)[1]. The incidence of NTDs is ~1 in 1000, but in some geographical regions, it is estimated to reach 4–10 in 1000[2,3].

NTDs are influenced by multiple genetic and environmental factors[1]. Results from recent studies have suggested that the abnormal homocysteine (Hcy) metabolism is related to the occurrence of NTDs[4]. Epidemiological studies reveal that abnormal maternal homocysteine during pregnancy is associated with an increased risk of NTDs in offspring[5,6]. Hcy is an intermediate of methionine metabolism, and homocysteine thyolactone (HTL), a metabolite of Hcy can directly modify proteins to affect their function. Hcy modification of albumin in human blood has been shown to influence its structure and function, and subsequently lead to disease development[7]. Furthermore, accumulating evidence indicate that human serum proteins are modified by Hcy, and the level of modification is regulated by the serum Hcy level[8]. Results from two other studies have demonstrated the presence of histone homocysteinylation in HEK293T[9] and human endothelial cells[10], however, there have been no published report on histone homocysteinylation in human fetal tissue.

Emerging evidence suggests an important role of histone marks in epigenetic metabolic control[11,12]. Because histone-modifying enzymes consume key metabolites, it is conceivable that they interpret the metabolic state of a given cell by changing chromatin modification patterns. Consistent with this, a global reduction of nuclear acetyl-CoA levels decreases histone acetylation, whereas reduced levels of $NAD^+$ have the opposite effect, inhibiting histone deacetylation[13,14]. An array of histone modifications including phosphorylation, acetylation, methylation, ubiquitination, and glycosylation have been shown to be associated with changes in chromatin organization, gene activation, silencing, and several other nuclear functions[15,16]. These findings suggest that Hcy could act as a substrate to modify histones and aberrant histone homocysteinylation may be involved in the failure of NTC.

Building on these previous observations, here we describe the identification and confirmation of homocysteinylation as a histone modification (histone KHcy). The level of histone KHcy was substantially increased under a high-Hcy environment in cells. We demonstrated that in neural stem cells, increased KHcy on histone H3 lysine 79 (H3K79Hcy) was associated with the downregulation of genes related to NTC. Our findings show histone KHcy is a previously unidentified histone modification. Our results suggest that high Hcy levels may increase the expression level of histone H3K79Hcy, resulting in a decrease in expression of some NTC-related genes, which lead to NTDs formation.

## Results

**Identification and verification of histone Hcy modification**. As the first step, the presence of histone homocysteinylation sites were analyzed on trypsin-digested core histones from 10 normal controls of human embryonic brain samples using HPLC/MS/MS (see Supplementary Table 1 for a summary of the information on the individual fetus). A mass shift of +174.04600 Da (monoisotopic mass) at lysine residues of the digested histone peptides was detected. Such a mass shift resembled what had been previously published, indicative of possible homocysteinylation[7]. A typical example of a lysine homocysteinylation-modified peptide, (H4K44Hcy), identified by MS, is shown in Fig. 1a. Within this peptide, a series of b- and y-type homocysteinylation fragment ions were evident which not only provided reliable sequence

information, but also indicated an unambiguous +174.04600 Da shift. For the simplicity purpose, we named the +174.04600 Da shift on histone lysine as histone lysine homocysteinylation (KHcy). Altogether, 39 histone KHcy sites were identified in four major histone variants from 10 normal controls (Fig. 1b and Supplementary Data 1), suggesting that homocysteinylation is a relatively common histone mark. Analysis on the frequency of each of the 39 KHcy sites across the 10 human embryonic brain samples revealed that while a number of KHcy sites were conserved among different samples, others were present only in one sample (detailed in Supplementary Table 2). In addition, most all of the identified histone KHcy sites are the sites that have been shown to be subject to other types of modifications[17], including H3K27, H3K36, and H3K79 whose modification is important for chromatin structure and function[18].

To further confirm the presence of KHcy in histones, we generated a rabbit polyclonal antibody against KHcy using homocysteinylated bovine serum albumin (BSA). Figure 1c shows that the antibody can detect a much stronger KHcy signal to homocysteinylated BSA than that against unmodified BSA. Furthermore, pre-incubation of homocysteinylated lysine with the antibody diminished the signal against homocysteinylated BSA, but had minimal effect on signal against unmodified BSA (Fig. 1c). In a separate line of experiments, the anti-KHcy antibody was found to specifically recognize only the homocysteinylated ovalbumin (OVA), but not the acetylated or succinylated OVA (Fig. 1d).

To confirm the presence of KHcy in histones, western blotting analysis was performed using an antibody against lysine homocysteinylation (anti-KHcy). A significant KHcy signal was observed in histone H3 from human embryonic brain tissues (Fig. 1e). Histone KHcy signal was also detected in samples from other human embryonic tissues including spinal cord, heart, liver, lung, kidney, muscle, and skin, although to a lesser extent (Fig. 1e). To further determine whether the KHcy is present in a broad range of species, we performed western blotting analysis using KHcy specific antibody on samples from *D. melanogaster*, Zebrafish, *Gallus gallus*, *M. musculus* to *H. sapiens*, and our results showed that KHcy signal could be detected in all samples (Fig. 1f). These findings suggest that the histone KHcy is an evolutionarily conserved modification among a wide range of species.

**The level of histone KHcy is regulated by HTL level in vitro**. Homocysteine thiolactone (HTL) is an intermediate metabolite in the metabolic pathway of Hcy, which has been reported to react with the ε-amino group of lysine residue in proteins (Fig. 2a)[19]. We performed an in vitro experiment to investigate whether histone homocysteinylation is the result of direct HTL modification, using histones purified from prokaryotic expression. Incubation of purified, unmodified histones H3, H4, H2a, and H2b with 5 mM HTL for 2 h led to significant homocysteinylation of all histones as revealed in dot blot analysis using anti-KHcy antibody (Fig. 2b, top panel). KHcy modification of histone H3 was found to be dose-dependent, i.e., increase in HTL treatment concentration resulted in an elevation in KHcy signal intensity (Fig. 2b, bottom panel). In addition, significant KHcy modification of histone H3 was evident after HTL treatment for 14 h (Fig. 2b, middle panel).

The levels of homocysteinylation of the aforementioned histone H3 were further evaluated because H3K4me1 or H3K27ac play a key role during differentiation of human embryonic stem cells to neuroepithelium[20], while H3K9me2 is found to be participating in neuronal differentiation, and H3K9 methylation and H3K4 methylation are involved in the nervous

system disease[21–23]. The results showed that the undigested H3 displayed a major peak of about 15KD, in accordance with the molecular weight of unmodified H3 (Fig. 2c, top panel). The HTL-treated histone H3 had one additional major peak with molecular mass greater than 15KD, and the difference in molecular mass between the adjacent two peaks is in the

proximity of 3 Hcy modifications, indicating that multiple, simultaneous KHcy modifications may exist on histone H3 during HTL treatment.

We were interested in defining possible sites of histone modification under HTL treatment, therefore we performed QE-HF mass spectrometry analysis on HTL-treated (5 mM, 2 h)

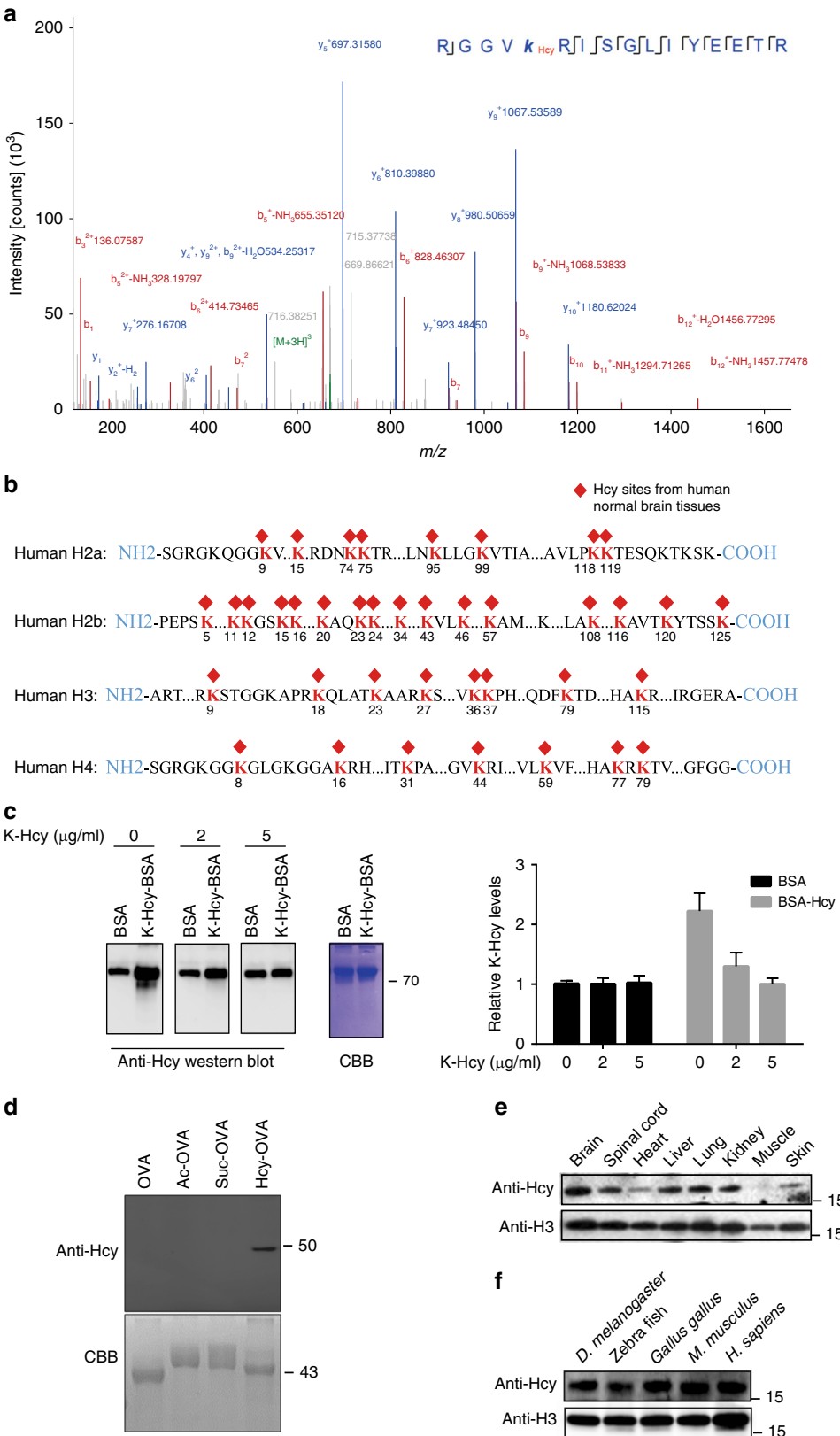

histones including H2a, H2b, H3, and H4. Figure 2d shows a typical MS/MS spectrum of H4K59 peptide (GVLK$_{Hcy}$VFLEN-VIR) identified from an HTL-treated sample. A mass shift of +174.04600 Da (monoisotopic mass) detected by mass spectrum indicated that there was an Hcy modification on the H4K59 peptide (Fig. 2d). A total of 24 histone KHcy-modified sites were found in all four histones (Fig. 2e and Supplementary Data 8), out of 57 lysine residues in all histones. The highest number of modification sites was found in H2b, while the least number of modification sites was present in H3 (Fig. 2e). It is worth mentioning that the number of homocysteinylation sites seemed to correlate with the intensity of KHcy signal (except for H4) on western blotting (Fig. 2b top panel and 2e).

Interestingly, 19 histone KHcy sites were also found in normal human fetal brain samples (depicted with red dot in Fig. 2e). It not only supports the data from HTL treatment, but also suggests that these same homocysteinylation histone sites including H3K79Hcy may be more exposed at the surface of these histones.

**The level of histone KHcy is regulated by cellular Hcy.** Since we observed that in vitro HTL treatment resulted in histone homocysteinylation (Fig. 2b, c), we set to evaluate whether cellular levels of HTL and Hcy could influence levels of histone KHcy. Inside cells, Hcy can be converted to HTL under catalysis of the cellular enzyme MetRS[24]. Figure 3a is a schematic diagram of the relationship between Hcy, HTL, and protein homocysteinylation. In our experiment, mouse neural stem cell line, NE4C (ATCC CRL-2925) was used. Cultured NE4C cells were treated with 0.1, 0.5, and 1 mM Hcy. Western blotting analysis of extracted histones revealed that histone homocysteinylation levels increased with increasing concentration of Hcy (Fig. 3b). A more profound dose-dependent histone homocysteinylation was also observed with increasing concentrations of HTL during the treatment of NE4C cells (Fig. 3c), suggesting a more direct effect of HTL dose on histone homocysteinylation. Treatment of HTL and Hcy also resulted in elevation of cellular histone homocysteinylation in HEK293 cells (Supplementary Figure 1A and Supplementary Figure 1B). Furthermore, knockdown of MetRS in HEK293T cells led to a reduction of endogenous HTL, leading to a reduced level of histone homocysteinylation (Supplementary Figure 1E). Our data provide evidence supporting the hypothesis that cellular metabolites of the homocysteine metabolism pathway may affect histone homocysteinylation.

To further corroborate these observations, label-free quantitative mass spectrometry (PRM: parallel Reaction Monitoring) was used to identify histone sites before and after pretreatment of HTL. 6 histone KHcy sites were detected in samples from untreated cells while as many as 20 of such sites were identified in cells treated with HTL (Fig. 3d and Supplementary Table 3).

These data clearly support the notion that cellular histone sites could be modified with increased HTL or Hcy level. Interestingly, we found that 18 of 20 histone KHcy sites in NE4C cells match the sites identified in human fetal brain (Fig. 3d), indicating that NE4C is an appropriate cell model to investigate the role of histone homocysteinylation in human NTDs formation.

**Validation of histone H3K79Hcy and its regulation by cellular Hcy.** H3K79 methylation plays important roles in embryonic development[25] and abnormal H3K79 dimethylation results in altered expression of a number of NTC genes and may be involved in NTDs[26]. In this study, results from mass spectrometry demonstrated that histone H3K79Hcy was enriched in all of human fetal brain tissue (Fig. 1b), HTL-treated commercial histone H3 (Fig. 2e), and cultured NE4C cells (Fig. 3d). Therefore, it is of significant rationale to investigate whether aberrant histone H3K79Hcy plays a role in the failure of NTC. Figure 4a shows a typical lysine H3K79Hcy-modified peptide, EIAQDFK$_{Hcy}$TDLR, from NE4C mass spectrometry data; a series of b- and y-type homocysteinylation fragment ions provided reliable sequence information and revealed the unambiguous homocysteinylation on histone H3K79.

To further verify the site of H3K79Hcy, an antibody against H3K79Hcy (anti-H3K79Hcy) was generated in our laboratory. Supplementary Figure 2A, 2B and 2C show that anti-H3K79Hcy antibody specifically recognizes homocysteinylated H3K79 (detailed in Methods). Initial western blotting studies using anti-H3K79Hcy revealed that H3K79Hcy was highly enriched in HeLa, HEK293, mouse brain and human fetal brain (Fig. 4b). Further analysis demonstrated that H3K79Hcy was widely expressed in almost all tissues in human fetus including brain, heart, liver, lung, kidney, spinal cord, muscle, skin and placenta (Fig. 4c).

To investigate whether the metabolite levels of intracellular Hcy was the driving force for H3K79Hcy modification, we performed an western blotting analysis with H3K79Hcy antibody on HTL-treated NE4C cells (Fig. 4d). A markedly increase in H3K79Hcy was observed with an increase dose of cellular HTL. In addition, treatment with HTL in cultured NE4C cells had an effect on histone methylation and acetylation as well (Fig. 4d).

The level of H3K79Hcy was also quantified using a mass spectrometry label-free (PRM) method and Skyline software. As shown in Fig. 4e, f, the modified peptides EIAQDFK$_{Hcy}$TDLR and IAQDFK$_{Hcy}$TDLR, which all contain H3K79Hcy modification, were readily detectable in samples from HTL treated NE4C cells but were not detectable in control group. Compared to H3K79Hcy, level of other histone modifications, i.e. methylation on H3K79 (EIAQDFK$_{me1}$TDLR) increased following 0.5 mM HTL treatment (Fig. 4g). Level of dimethylation on H3K79 (EIAQDFK$_{me2}$TDLR) did not change significantly following HTL treatment (Fig. 4h).

**Fig. 1** Histone homocysteinylation is a common modification among different tissues and species. **a** A typical HPLC-MS/MS spectra of a tryptic peptide 'RGGVK$_{Hcy}$RISGLIYEETR' harboring H4K44 homocystylation, derived from human brain. The x and y axes represent m/z and relative ion intensity, respectively. A series of b- and y-type homocysteinylation fragment ions are evident which not only provide reliable sequence information, but also indicate an unambiguous +174.04600 Da shift for Hcy. **b** Schematic illustration of homocysteinylation sites of histone lysine residues in human normal brain samples identified using HPLC-MS/MS. The red diamond shape depicts homocysteinylation sites in core histones (H3, H4, H2a, and H2b). The number underneath each red lysine residue (K) represents the position of the particular lysine residue within each respective histone. **c** Verification of anti-KHcy antibody. The homocysteinylation levels of BSA (Bovine Serum Album) and KHcy modified BSA were detected with anti-KHcy antibody under the presence of 0, 2, or 5 μg/ml of Hcy modified lysine (KHcy as competitor). CBB Coomassie Brilliant Blue staining. These test repeated for 3 times and the quantitation of the western blotting showed on right. In the BSA group, the relative K-Hcy levels were 1 ± 0.05, 1.15 ± 0.10; 1.11 ± 0.78. In the BSA-Hcy group, the relative K-Hcy levels were 10.88 ± 1.02, 5.48 ± 0.34; 1.39 ± 0.21. **d** Verification of specificity of the anti-K-Hcy antibody. Western blotting assay was carried out by incubating the anti-KHcy antibody with unmodified OVA (ovalbumin), acetylated-OVA, succinylated-OVA, or K-Hcy-OVA. **e** Western blotting analysis for the detection of H3 homocystylations in samples from a variety of human fetal tissues, including brain, spinal cord, heart, liver, lung, kidney, muscle, and skin. Anti-Hcy: rabbit polyclonal anti-Hcy antibody; Anti-H3: rabbit polyclonal anti-H3 antibody. **f** Presence of H3 homocysteinylation in different species, including *D. melanogaster*, Zebra fish, *Gallus gallus* brain, mouse brain, and human fetal brain, demonstrated using western blotting with rabbit polyclonal anti-Hcy and anti-H3 antibodies

Taken together, our results provide sufficient evidence supporting histone H3K79Hcy. In addition, our results demonstrate that the level of histone H3K79Hcy is regulated by intracellular Hcy intermetabolites.

**Increase of NTDs and H3K79Hcy level in HTL treated chicken.** A number of previous have demonstrated that increases in levels of Hcy or HTL do not lead to NTDs in mice[27–29]. Therefore, we explored the potential link between high levels of HTL, as well as elevated levels of histone H3K79Hcy, and the failure of NTC in in vivo experiments using chicken model.

Chicken represents an appropriate animal model to analyze dynamics of neurulation, and has advantages over other models, including a short period of embryogenesis and low cost[30]. The expression pattern of NTC genes is similar in chicken and human embryo, derived from a conservation in chromosomal localization of these genes[31].

After incubation for 28–30 h, single injection of 0.5 μl of 0.5 mM HTL was carried out into the neural groove of chicken embryos, and chicken embryo malformations of all organs were evaluated on Embryonic Day 5 (E5). In the control group, the chicken embryo survival rate was 95.83% (46/48) with a

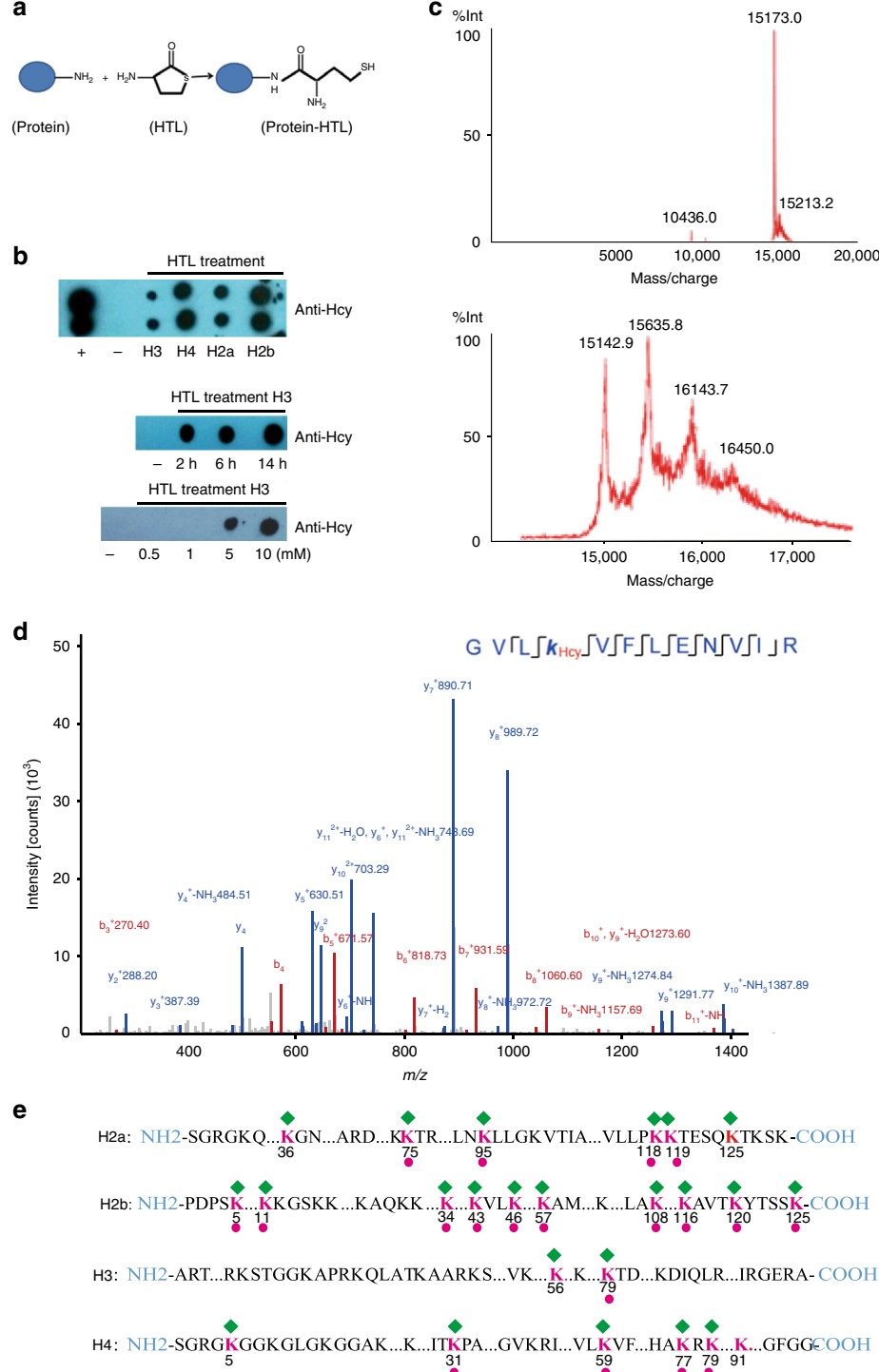

**Fig. 2** Direct in vitro histone homocysteinylation by HTL. **a** Schematic illustration of protein modification by HTL. **b** Dot-blot analysis of histone homocysteinylation by HTL. The unmodified histones H3, H4, H2a, and H2b expressed from *E. coli* were used. Top panel: four histones were incubated with 5 mM HTL for 2 h and histone homocysteinylation was detected using anti-Hcy antibodies. (+: positive control, tubulin antibody diluted 1:1000 was used as the positive control; –: negative control, sodium phosphate buffer was used as the negative control); Middle panel: histone H3 was treated with 5 mM HTL for 2, 6, and 14 h, respectively, and histone homocysteinylation was detected using anti-Hcy antibodies. Bottom panel: histone H3 was treated with 0.5 mM, 1 mM, 5 mM, and 10 mM HTL respectively for 2 h and histone homocysteinylation was detected using anti-Hcy antibodies. **c** MALDI analysis of unmodified H3 from *E. coli* with (bottom) or without (top) in vitro HTL treatment. The undigested H3 display a major peak of about 15KD. Additional major peaks greater than 15KD are seen in HTL-treated H3 samples. The difference in molecular mass between the adjacent two peaks is in the proximity of 3 Hcy modifications, indicating that multiple, simultaneous KHcy modifications may exist on H3 during HTL treatment. The *x* and *y* axes represent *m/z* and relative ion intensity, respectively. **d** A typical HPLC-MS/MS spectra of a tryptic peptide 'GVLK$_{Hcy}$VFLENVIR' derived from HTL-treated H4 with homocystylation at H4K59 site. The *x* and *y* axes represent *m/z* and relative ion intensity, respectively. A series of b- and y-type homocysteinylation fragment ions are evident which not only provide reliable sequence information, but also indicate an unambiguous +174.04600 Da shift for Hcy. **e** Illustration of histone homocysteinylation sites identified by HPLC-MS/MS analysis on unmodified core histones treated with HTL. The green diamond shape depicts homocysteinylation sites in core histones (H3, H4, H2a, and H2b). The number underneath each red lysine residue (K) represents the position of the particular lysine residue within each respective histone. Homocysteinylation sites, present both naturally in normal human brain samples (Fig. 1b) and after in vitro HTL treatment are marked with a red dot

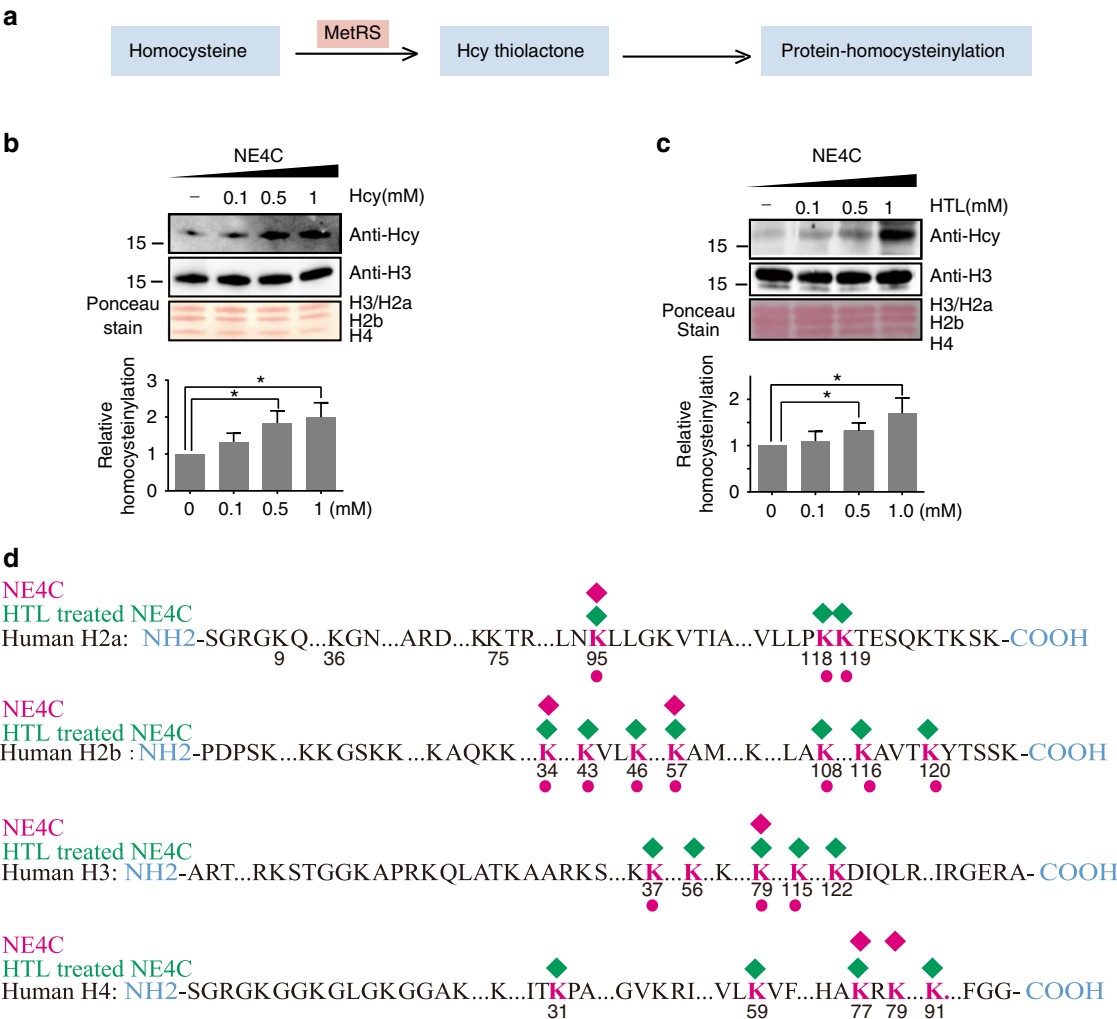

**Fig. 3** Intracellular histones homocysteinylation. **a** Schematic diagram of the process for intracellular protein homocysteinylation by Hcy. **b, c** Western blotting analysis of H3 homocysteinylation in NE4C cells treated with increasing doses of Hcy (**b**) or HTL (**c**) for 4 h. Data represent mean ± SEM (*n* = 3). *$p<0.05$ vs. 0 mM Hcy (**b** bottom chart) or HTL (**c** bottom chart); the *p* values were calculated with unpaired *t* test. **d** Schematic illustration of histone homocysteinylation sites identified from NE4C treated with or without 0.5 mM HTL for 4 h using HPLC-MS/MS. The detected homocysteinylation sites from treated or untreated samples are shown in green or red diamond shape, respectively. The number underneath each red lysine residue (K) represents the position of the particular lysine residue within each respective histone. Homocysteinylation sites detected in both normal fetal brain (Fig. 1b) and NE4C are marked with red dots under numbering of lysine residues

malformation rate was 2.08%, and the only malformation was NTDs. In the group injected with 0.5 µl of 0.5 mM HTL, the chicken embryo survival rate was 67% (37/55) and the malformation rate was 43.63% (24/55). The malformations included NTDs, heart defects, brain atrophy and tail deformity. Among them, 20 chicken embryos showed NTDs. A typical open spina bifida phenotype and a meningeal encephalocele phenotype

of embryo 8 (E8) were shown (Fig. 5a, b), while Fig. 5c showed a normal control on E8.

Western blotting assay was performed to compare H3K79Hcy levels in samples from the control and the HTL-treated group. Higher levels of histone H3K79Hcy were detected in samples from chickens of HTL-treated group with phenotypes of NTDs (Fig. 5d). To further explore the possible role of H3K79Hcy

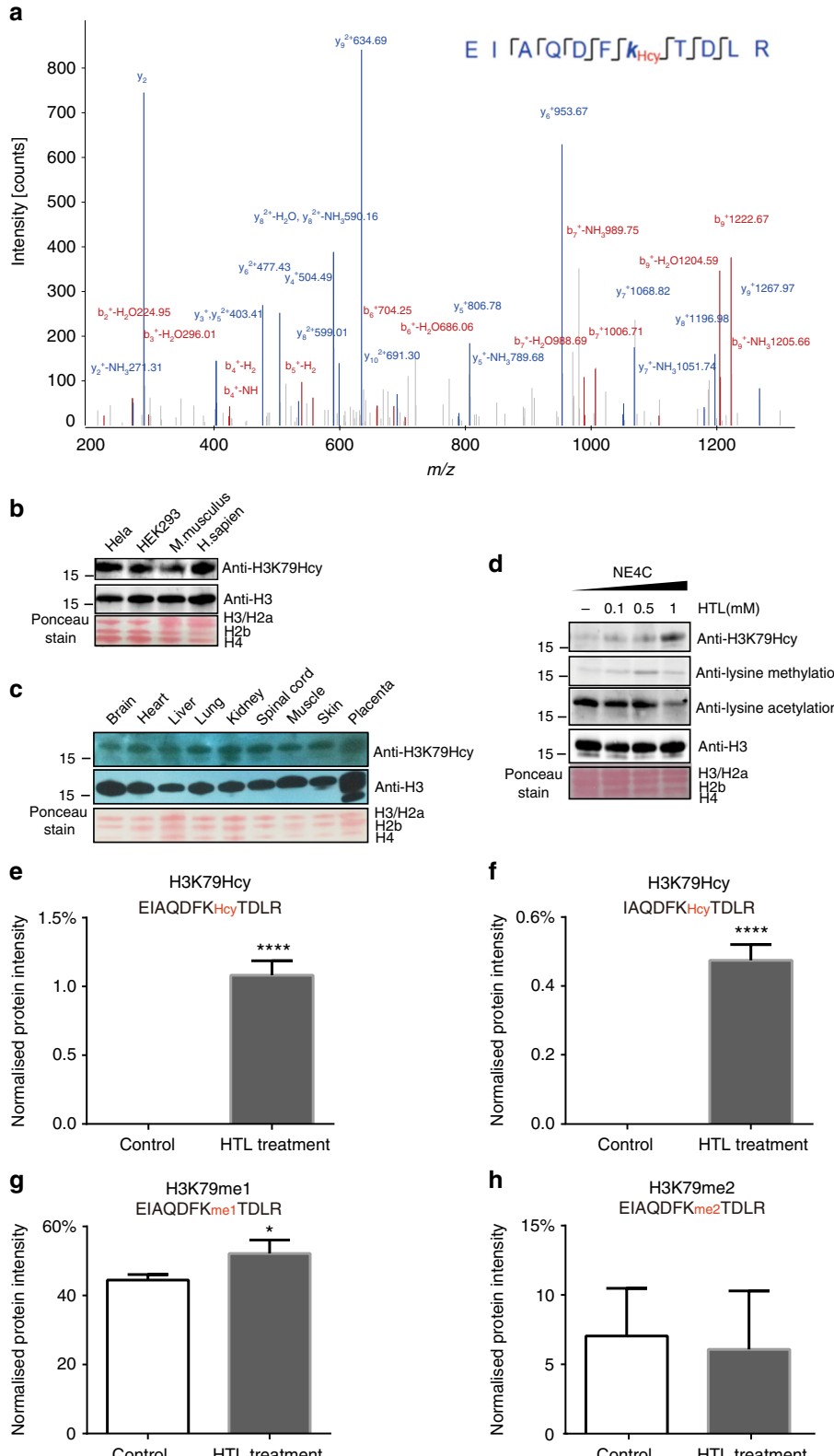

**Fig. 4** Histone H3K79Hcy validation and regulation by cellular Hcy level. **a** A typical HPLC-MS/MS spectra of a tryptic peptide 'EIAQDFK$_{Hcy}$TDLR' including H3K79 homocystylation derived from NE4C cells. The x and y axes represent m/z and relative ion intensity, respectively. **b–d** Western blotting analysis of H3K79Hcy modification. **b** In different species including mouse brain, Hela cell, HEK293 cell, and human brain; **c** In samples from a variety of human tissues, including brain, heart, liver, lung, kidney, spinal cord, muscle, skin, and placenta. **d** Cell lysates from NE4C treated with different concentrations. Anti-H3K79Hcy rabbit polyclonal anti-H3K79; Anti-H3: rabbit polyclonal anti-H3 antibody. Two additional antibodies, anti-lysine methylation and anti-lysine acetylation were included in **d**. Ponceau stain was used to show consistency of protein loading in each lane. **e–h** Quantitation of targeted H3K79 peptides by PRM MS method and skyline analysis software: EIAQDFK$_{Hcy}$TDLR (**e**), IAQDFK$_{Hcy}$TDLR (**f**), EIAQDFK$_{me1}$TDLR (**g**), and EIAQDFK$_{me2}$TDLR (**h**) in control and 0.5 mM HTL treated NE4C, using PD and skyline software analysis. Data represent mean ± SEM (n = 3). *p<0.05, ****p<0.0001 vs. control; the p values were calculated with unpaired t test

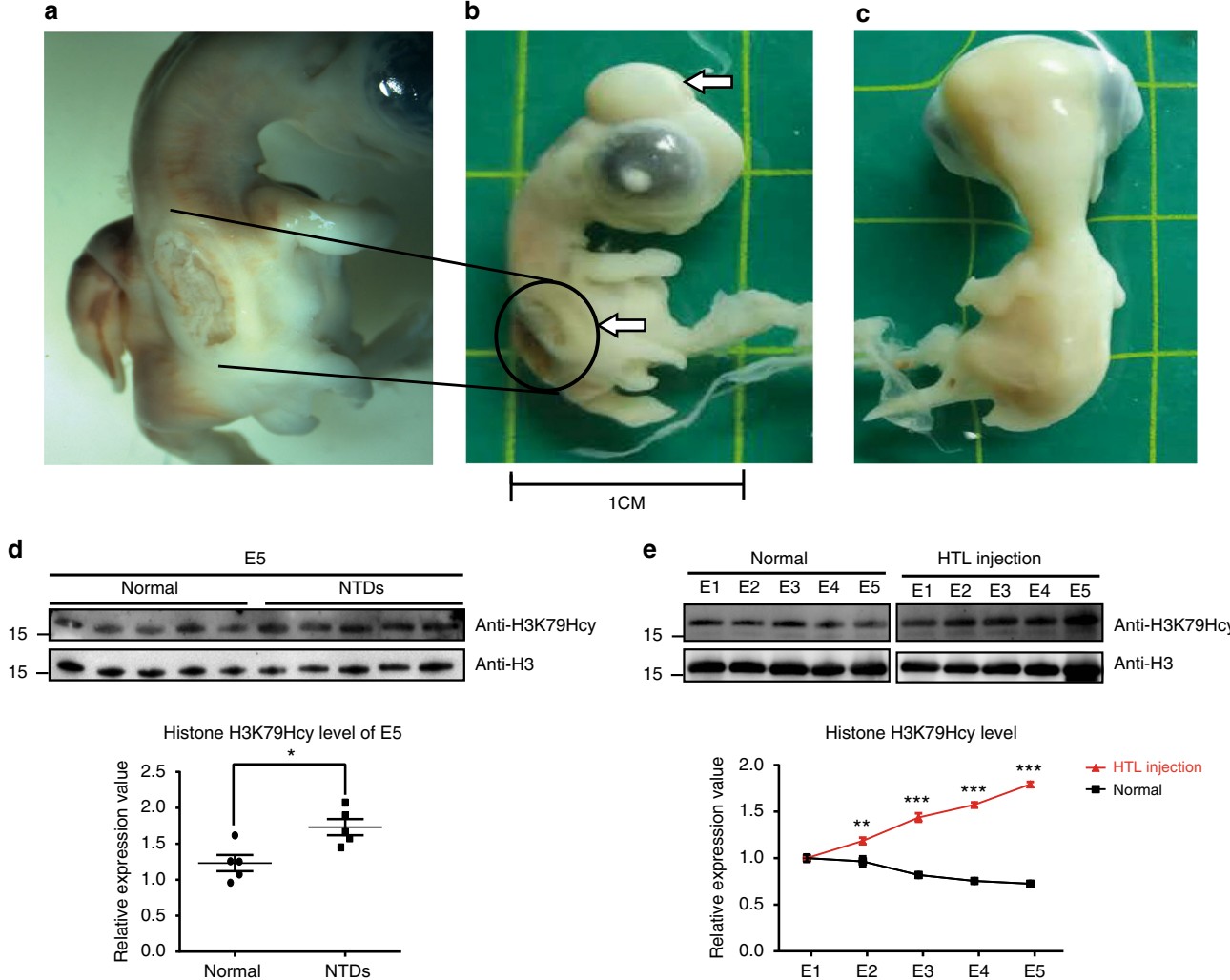

**Fig. 5** Increase of histone H3K79Hcy in high-HTL-induced chicken NTDs. **a–c** Eggs treated with 0.5 mM-HTL exhibited NTDs phenotypes at stage of embryo 8 (E8); the arrow in **b** (upper) showed a meningoencephalocele and the arrow in **b** (below) showed spinal bifida aperta. **a** is a local enlargement of spina bifida aperta. **c** is the normal phenotypes of chicken at stage of embryo 8 without HTL treatment. **d** Increased expression of histone H3K79Hcy in high-HTL-induced chicken NTDs. Upper panel: western blotting of H3K79Hcy modification from normal (n = 5) and high-HTL-induced chicken NTDs (n = 5); lower panel: quantification of the western blotting signal intensity shown in scatter plot. Normal: 1.232 ± 0.1117 (n = 5); NTDs: 1.732 ± 0.1119 (n = 5); *p = 0.0133, NTDs vs. normal brain tissues. The p values were calculated with unpaired t test. **e** Increase in histone H3K79Hcy expression during brain development in high-HTL-treated chicken and decrease in histone H3K79Hcy expression during normal chicken development. E1: 24 h after embryos incubation; E2: 48 h after embryos incubation; E3: 72 h after embryos incubation; E4: 96 h after embryos incubation; E5: 120 h after embryos incubation. Data represent mean ± SEM (n = 3). **p<0.01, ***p<0.001 vs. control; the p values were calculated with unpaired t test

during brain development, we compared levels of H3K79Hcy from E1 to E5 between normal group and HTL injection group (Fig. 5e). The results demonstrated an increase in histone H3K79Hcy expression during brain development in high-HTL-treated chickens and a decrease in histone H3K79Hcy expression during normal chicken development (Fig. 5e), indicating that abnormal H3K79Hcy expression may lead to the occurrence of NTDs in chicken. Our data suggest that elevated H3K79Hcy modification may underlie the failure of NTC during early development due to functional disturbance of Hcy metabolism.

**Genomic localization of histone H3K79Hcy**. To further explore the importance of histone H3K79Hcy during neural system development, in vitro ChIP-seq analysis was performed with NE4C cells using anti-H3K79KHcy antibody. H3K4me3, which is highly enriched in promoter regions, was used as a positive control[32]. A total of 8197 peaks from 3255 genes were detected using anti-H3K4me3 antibody, while 7299 peaks from 1277 genes were identified using anti-H3K79Hcy antibody, scanning through the entire mouse genome (Fig. 6a, b; Supplementary Data 2, 3 and 4). MAnorm was employed to compare peak regions enriched by anti-H3K79Hcy and anti-H3K4me3 antibodies, and ChIP-seq common peaks were used as reference to build the rescaling model of normalization[33]. Profound differences were observed in patterns of enriched peaks between ChIP-seq data obtained with the two different antibodies (Supplementary Figure 3A), suggesting that histone H3K79Hcy may have a function other than H3K4me3. Supplementary Figure 3B illustrates that using three different antibodies, a number of peaks were identified by ChIP within the region containing the *Smurf2* gene. These data support our notion that both H3K4me3 and H3K79Hcy antibodies bind their respective targets effectively during ChIP-seq assay, although genomic location of the targets and binding efficiency may vary.

ChIP Gene Ontology term analysis showed that, among the genes targeted with H3K79Hcy, there was bias favoring nervous system-related genes. In the top 10 GO groups of genes with enriched peaks, the top 4 are related to the nervous system (Table 1). The group of genes with the most enriched peak groups were identified as those involved in nervous system development, followed by genes involved in the generation of neurons and neurogenesis. We then used the DAVID method[34] to perform functional annotation clustering for the biological processes of genes with H3K79Hcy peaks and the results are shown in Supplementary Data 5. And the top 3 enriched functional cluster of genes was found to be associated with neuron differentiation, neuron migration and regulation of nervous system development. The network of 6 interesting functional clusters was generated by FGNet[35] from Bioconductor project (Fig. 6c). It also shows that most of the H3K79Hcy-regulated genes were associated with neurodevelopment.

The next question that arises in this context is whether H3K79Hcy can regulate the expression level of genes with enriched peak in ChIP-seq assays. We compared the H3K79Hcy enrichment level to the gene body obtained from ChIP-seq (Supplementary Data 6) and the expression level of these H3K79Hcy binding genes define in RNA-seq analysis (Supplementary Data 7). All genes were divided into five groups according to their enrichment level at each 20% percentile. and the expression level of each gene within each group was analyzed. Our results showed that gene expression was gradually elevated with the increase of H3K79Hcy enrichment level (Fig. 6d), while H3K4me3 enrichment in the promoter was associated with gene activation, which is consistent with findings from previous studies (Fig. 6e)[36].

**Enrichment and expression levels of NTC related genes**. We were interested in investigating if and how the level of histone H3K79Hcy might affect H3K79Hcy binding to NTC related genes and subsequently, their expression. Among over 300 genes and 14 epigenetic regulator genes connected with NTDs in the mouse, only *Cecr2*, *Smarca4*, and *Dnmt3b* are founded in H3K79Hcy peak genes. Therefore, we focused our first set of experiments on *Smarca4*, *Cecr2*, and *Dnmt3b*. We analyzed H3K79Hcy enrichment on *Smarca4*, *Cecr2*, and *Dnmt3b* in NE4C cells under normal or HTL treatment conditions. *Smarca4*, *Cecr2*, and

*Dnmt3b* are NTC related genes, the loss of function of which has been shown to result in NTDs.

ChIP-seq analysis showed that the H3K79Hcy enrichment levels on the three NTC genes were all higher in untreated NE4C than in HTL-treated NE4C (Fig. 7a, top panel). The most profound effect of HTL-treatment on H3K79Hcy enrichment level was observed in *Cecr2* genes. Further experimentation using ChIP-qPCR confirmed that in HTL-treated NE4C cells, H3K79Hcy enrichment of *Smarca4*, *Cecr2*, and *Dnmt3b* were decreased (Fig. 7b).

Next, we evaluated the level of H3K79Hcy enrichment in different regions of *Smarca4*. As shown in Fig. 7c, in untreated NE4C cells, H3K79Hcy enrichment was found to be significantly higher within the *Smarca4* gene body than the upstream and downstream regions of this gene. Upon HTL treatment profound reduction of H3K79Hcy enrichment was evident in the gene body of *Smarca4* than the other two regions, implying that the increase in H3K79Hcy level may hinder the binding of H3K79Hcy to its targets.

Lastly, the potential effect of HTL-treatment and diminished H3K79Hcy binding to these three genes on their respective gene expression was investigated. Not surprisingly, results from RNA-seq analysis indicated a diminished expression of these three genes in HTL-treated NE4C (Fig. 7a, below panel). Further RT-PCR confirmed that the expression level of these three genes decreased in HTL-treated NE4C (Fig. 7d).

In addition, we also found that some Smarca4-regulated genes, which were associated with NTDs, exhibited decreased expression upon HTL treatment, i.e. SHH signaling pathways and their downstream target genes (Supplementary Figure 4A)[37]; PCP signaling pathway-related genes (Supplementary Figure 4B)[38]; and self-renewal/proliferation genes (Supplementary Figure 4C)[39]. Meanwhile, the expression of the housekeeping genes *Gapdh* and *Actg1* was unchanged (Supplementary Figure 4D). These results indicated that decreased expression of *Smarca4* in turn led to the decrease of some NTC-related genes and pathways to play a role in NTDs formation.

Collectively, our data from this study indicate that during HTL-treatment, the H3K79Hcy enrichment on *Smarca4*, *Cecr2*, and *Dnmt3b* was decreased, accompanied by decreases in their expression level while the overall level of histone H3K79Hcy was increased. These results suggest that H3K79Hcy is critical for *Smarca4*, *Cecr2*, and *Dnmt3b* expression.

**Increase of H3K79Hcy with decreased expression of NTC genes**. Knock-out of NTC-related genes leads to NTDs phenotypes in mice, indicating that suppression of the transcription of these genes, including *Smarca4*, *Cecr2* and *Dnmt3b*, is functionally connected to the pathogenesis of NTDs[40–42]. Because of the observed alteration of the transcription levels of these NTC genes under aberrant H3K79Hcy, we reasoned that aberrant H3K79Hcy might also have detrimental consequences in humans including the formation of NTDs. To test this hypothesis, we first measured Hcy levels in brain tissue samples from 10 normal fetuses and 10 NTDs cases (see Supplementary Table 1 for a summary of the information on the individual fetuses). As shown in left chart of Fig. 7e, brain Hcy level was significantly higher in samples from NTDs cases (41.0 pmol/mg), compared to 3.3 pmol/mg in normal controls. Using the anti-H3K79Hcy antibody, western blotting analysis was performed to evaluate H3K79Hcy levels in these samples. Stronger H3K79Hcy signals were detected in NTDs samples, compared to that in controls (Fig. 7e, middle chart). Average level of H3K79Hcy expression normalized to H3 was found to be significantly higher in NTDs samples (0.44 vs. 0.30 in controls, $p = 0.024$; Fig. 7e, right chart).

Along with the elevation in Hcy and H3K79Hcy levels in these NTDs tissues, there was a repressed transcription of the above-mentioned NTC-related genes. Results from nanostring analysis revealed an evident reduction in average levels of the transcription of these three genes (Fig. 7f).

Taken together, our data indicate that high Hcy levels in NTDs may result in an increase in the level of histone H3K79Hcy which may have a suppressive effect on the transcription of *Smarca4*, *Cecr2*, and *Dnmt3b* genes, leading to the failure of NTC.

## Discussion

Abnormal Hcy metabolism has been implicated in the occurrence of NTDs in a number of studies. Elevated maternal Hcy during pregnancy has been found to be associated with an increased risk

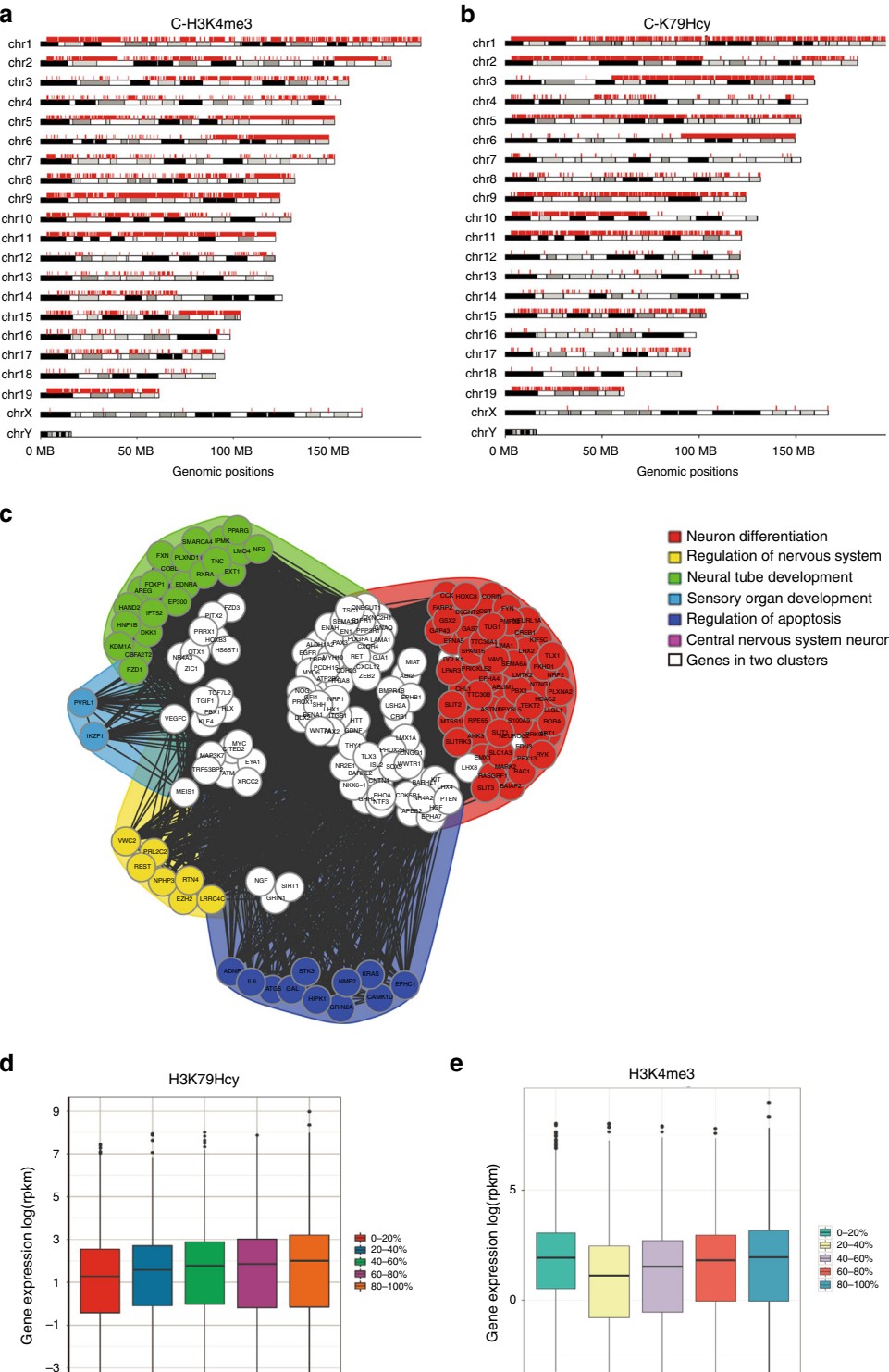

**Fig. 6** Enrichment of histone Hcy and H3K79Hcy on chromatin. **a**, **b** Genome-wide ChIP-seq analysis of peaks for histone H3K4me3 (**a**) or H3K79Hcy (**b**) in chromatin from NE4C cells. The peak was called by SICER, and the distribution of peaks was plotted by gtrellis. The red bars represent loci where the peaks located. **c** Functional network of enriched genes with H3K79Hcy peaks. DAVID method was used to do functional annotation clustering for biological process annotations of genes with H3K79Hcy peaks. The network of 6 functional clusters was generated by FGNet from Bioconductor project including genes of neuron differentiation genes (RED); genes of regulation of neurogenesis (YELLOW); genes of neural tube development (GREEN); genes of sensory organ development (LIGHT BLUE); genes of regulation of apoptosis (BLUE); genes of central nervous systerm neuron (PURPLE) and the genes in two clusters (WHITE). **d** Correlation between ChIP density in gene body and the level of gene expression. All genes were arbitrarily divided into 5 groups based on their H3K79Hcy ChIP density in gene body. The expression level of each gene was analyzed using RNA-seq. The $y$-axis represents the $\log^2$ (RPKM) value. 0–20%: median, 1.17; min, 0; max, 7.63; 25% quantile, −0.67; 75% quantile, 2.53; 20–40%: median, 1.48; min, 0; max, 8.01; 25% quantile, −0.31; 75% quantile, 2.64; 40–60%: median, 1.66; min, 0; max, 8.00; 25% quantile, −0.33; 75% quantile, 2.85; 60-80%: median, 1.71; min, 0; max, 7.98; 25% quantile, −0.34; 75% quantile, 2.96. 80–100%: median, 2.01; min, 0; max, 8.97; 25% quantile, −0.67; 75% quantile, 3.23. **e** Correlation between ChIP density in promoter and the levels of gene expression. All genes were arbitrarily divided into 5 groups based on their H3K4me3 ChIP density in promoter. The expression level of each gene was analyzed using RNA-seq. And value. 0–20%: median, 1.87; min, 0; max, 8.35; 25% quantile, 0.36; 75% quantile, 3.05; 20-40%: median, 1.02; min, 0; max, 8.01; 25% quantile, −1.14; 75% quantile, 2.40; 40–60%: median, 1.42; min, 0; max, 7.91; 25% quantile, −0.82; 75% quantile, 2.67; 60–80%: median, 1.74; min, 0; max, 7.80; 25% quantile, −0.33; 75% quantile, 2.93. 80–100%: median, 1.86; min, 0; max, 8.97; 25% quantile, −0.31; 75% quantile, 3.14

| Table 1 Top 10 of modification binding genes analyzed by biological process Gene Ontology | | |
|---|---|---|
| | **H3K79Hcy** | **H3K4me3** |
| Top1 | Nervous system development | Regulation of cellular macromolecule biosynthetic process |
| Top2 | Generation of neurons | Regulation of RNA metabolic process |
| Top3 | Neurogenesis | Anatomical structure morphogenesis |
| Top4 | Neuron differentiation | Regulation of cellular metabolic process |
| Top5 | Multicellular organismal development | Regulation of metabolic process |
| Top6 | Cell projection organization | Regulation of nucleobase-containing compound metabolic process |
| Top7 | System development | Regulation of macromolecule biosynthetic process |
| Top8 | Single-organism developmental process | Central nervous system neuron differentiation |
| Top9 | Anatomical structure development | Regulation of RNA biosynthetic process |
| Top10 | Ion transport | Multicellular organismal development |

of NTDs in offspring[4–6,43–46]. In chicken embryos, applying Hcy and HTL supplementation during early stages of chicken embryo development lead to the onset of NTDs in the embryos[47,48]. Similar results have been produced in the present study (Fig.5). All these findings suggest that high maternal Hcy is associated with the occurrence of NTDs, and that Hcy accumulation is a risk factor for NTDs. However, the underlying pathological mechanisms have not been fully elucidated.

Increasing evidence has implicated altered histone modifications in translating cellular metabolic states into changes in gene expression[11,49]. Several lines of evidence have shown that there is a strong relationship between one-carbon metabolism nutrients and epigenetic phenomena[50,51]. A causal link between histone methylation and nutritional status has also been demonstrated in yeast and human cells, where folate and methionine deficiency are associated with a reduction of histone methylation, mainly H3K4 methylation, and lead to changes in gene expression. In the present study, we explored the pathways from one-carbon metabolism intermediate Hcy to the onset of NTDs based on several key observations, the demonstrated increase of maternal Hcy in women who give birth to infants with NTDs, our discovery of modifications in histone KHcy as well as H3K79Hcy, and well-established altered expression of NTC genes in NTDs.

Protein homocysteinylation has been reported for a number of proteins, mostly enzymes, and result in alteration of protein function[7,8,52]. Given the connection between maternal high Hcy levels and onset of NTDs, we reasoned that the presence of Hcy would serve as a substrate to modify histones and to regulate the expression level of some NTC-related genes. From human embryonic brain samples, we identified 39 KHcy modification sites (Fig. 1b). To our knowledge, this is the first time that homocysteinylation specific to histone in human fetal brain has

been reported. Using an anti-KHcy antibody, we further demonstrated that histone homocysteinylation is a common histone modification present not only in different organs in humans, but in different species as well (Fig.1c, d).

We then performed an in vitro experiment to treat histones from prokaryotic expression (devoid of any modification) with HTL and to define KHcy sites under these conditions (Fig. 2e). Furthermore, KHcy sites were also defined in NE4C cells under normal and HTL-treatment conditions (Fig. 3d). Our data provide evidence supporting the overall fidelity of HTL treatment in vitro or in vivo for histone homocysteinylation resemble naturally occurring KHcy sites detected in fetal brain tissue samples. Although histone KHcy sites increased while HTL were used during in vitro and in vivo treatment, the most histone KHcy sites (39 Histone KHcy sites) were detected from fetal brain samples. These indicated that in the brain, histone homocysteinylation involving cellular metabolism is far more efficient than direct chemical reactions with HTL or Hcy.

Among all histone KHcy modifications, histone H3K79Hcy was naturally present in untreated NE4C cells as well as fetal brain samples, suggesting that it might be one of the key regulators for histone KHcy modifications. To investigate the possible mechanism, we performed ChIP-seq analysis in NE4C, comparing patterns of gene binding between H3K79Hcy and H3K4me3, a well-known epigenetic regulator of gene expression. Combining ChIP-seq and RNA-seq data, we showed that H3K4me3 was more enriched in the promoter and the gene expression level increased gradually along with an increase in ChIP density in the promoter region (Fig. 3e), consistent with previous findings[53]. However, a significant enrichment of histone H3K79Hcy was found to be in the gene body region (Supplementary Figure 3C), and a bioinformatics analysis showed that

the gene expression level increased gradually along with an increase in ChIP density within the gene body, rather than the promoter (Fig. 6d and Supplementary Figure 3E). These results indicate that H3K79Hcy may regulate the expression of gene to which it bound through its enrichment in the gene body region.

We further explored the effect of H3K79Hcy on expression of NTC-related genes and focused our efforts on Smarca4, Cecr2, and Dnmt3b. Among 14 epigenetic regulators required for NTC (https://ntdwiki.wikispaces.com/Epigenetic+Regulators), these 3 genes have been identified to be regulated by H3K79Hcy,

knocking out each one of them in mice leads to NTDs[40–42]. Cecr2, a strain-specific modifier which has shown both a hypomorphic and a presumptive null mutation on two different backgrounds: one susceptible (BALB/c) and one resistant (FVB/N) to NTDs. Dnmt3b is essential for de novo methylation and for mouse development. Smarca4 null mice exhibit embryonic lethality, while Smarca4 heterozygous mice show developmental defects, among them, 30% are NTDs[40,54,55].

Smarca4 (also known as Brg1) is the essential ATPase subunit of the mammalian SWI/SNF chromatin remodeling complex, and

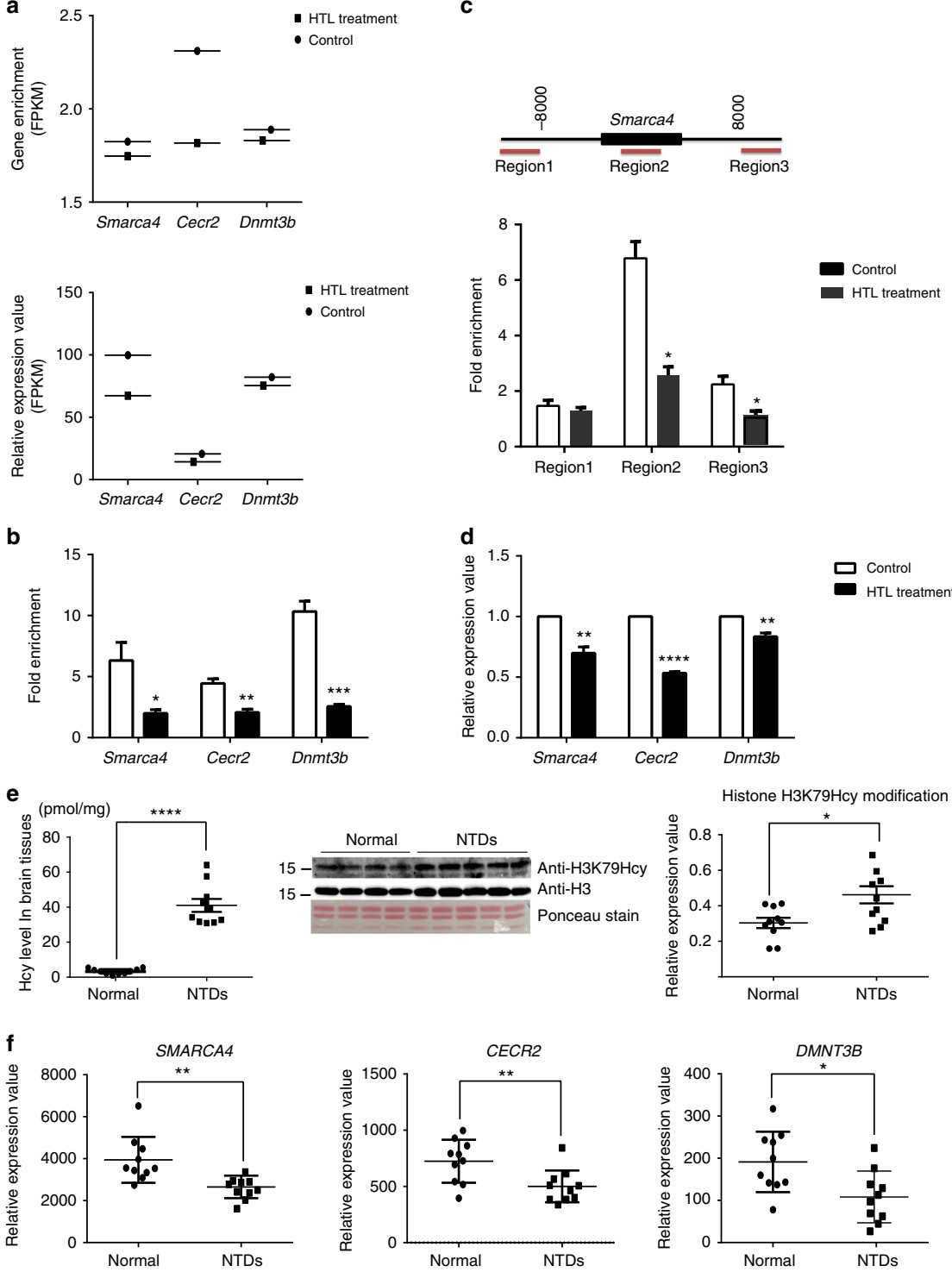

can alter the histone–DNA linkages in the target gene promoter region, slide the nucleosomes, expose the target gene promoter region to alter the specific transcription factor and its complexes for DNA accessibility, thereby regulating gene expression[56,57]. Our research also showed that some *Smarca4*-regulated genes, which are associated with NTDs, exhibited decreased expression upon HTL treatment, i.e. genes in the SHH signaling (Supplementary Figure 4A)[37]; PCP signaling (Supplementary Figure 4B)[38]; and self-renewal/proliferation genes (Supplementary Figure 4C)[39,54], suggesting that decreased expression of *Smarca4* plays a role in NTDs formation.

Binding of H3K79Hcy to these 3 genes has been verified in our studies, providing the basis for evaluating the effects of H3K79Hcy on its binding to these 3 genes and subsequent regulation on gene expression. Interestingly, in human NTDs samples where a higher level of Hcy was detected and an increase of overall H3K79Hcy was observed (Fig. 7e), the expression of above-mentioned 3 genes were all decreased (Fig. 7f). The answer to this dilemma lies in the results in Fig. 7 which demonstrates that there is a diminished specific binding of H3K79Hcy to these 3 genes upon HTL treatment (determined using ChIP-seq and ChIP-qPCR), leading to a decreased level of expression of these 3 genes (assayed using RNA-seq and qRT-PCR). Combining data from NE4C cells treated with HTL (Fig. 7a, b, d) and that from NTDs samples (Fig. 7e, f), suggests that higher cellular levels of HTL or Hcy confers to an elevated level of KHcy, in particular, H3K79Hcy. However, higher level of H3K79Hcy has a negative impact in its binding to aforementioned genes, resulting in a reduced gene expression of *Smarca4*, *Cecr2*, and *Dnmt3b*. Currently, we are conducting mechanistic studies to define differences in chromosomal structure, transcritomal regulations, as well as nucleosomal positioning in the context of H3K79Hcy.

It is possible that the Hcy level and histone homocysteinylation level presented in this study may not accurately reflect that of NTC at the time of neurulation because the fetuses used in this study were in at least the second trimester, long after neurulation, and NTD samples in this study were mostly spina bifida, which occurs at very early spina cord development. However, results from chicken embryo model showed that there was an increase in histone H3K79Hcy expression during brain development in high-HTL-treated chickens while a decreased expression of H3K79Hcy was detected in samples from the normal group, suggesting that abnormal H3K79Hcy expression might lead to the occurrence of NTDs in chicken.

Taken together, our findings presented in this study identify histone KHcy as a mechanism by which Hcy regulates cellular physiology and supports a model in which a shift in the cellular utilization of energy source alters gene expression in a metabolite-directed manner.

## Methods

**Human subjects details**. The NTDs and normal control samples were from the Lüliang area of Shanxi Province in northern China from March 2004. Fetuses with NTDs were from medical abortions and had been diagnosed with spina bifida by B-mode ultrasound in the early stages of pregnancy; the sex, gestational age and general development were also recorded in detail. The pathological diagnosis of NTDs was completed by experienced pathologists in accordance with the International Classification of Disease, Tenth Revision, codes Q00.0, Q05.9, and Q01.9 (http://apps.who.int/classifications/). Control fetuses that had been aborted for non-medical reasons were enrolled from the same region[46,58]. Any fetuses displaying pathological malformations or intrauterine growth retardation were excluded from the control group. In this study, 10 samples with the highest Hcy levels from 173 NTDs were selected, and 10 samples were selected from 178 controls. The controls were matched with gender (Female: 5–6 cases; male: 4–5 cases) and age (< 20w: 1–2 cases; 20–30w: 7 cases; > 30w: 1–2 cases). The information collected from questionnaire during patient enrollment indicates that none of the mothers from either the controls or the NTDs group had received any folic acid supplements (please see Supplementary Table 1 for detail). The investigation was approved by the Committee of Medical Ethics of the Capital Institute of Pediatrics. Written informed consent was obtained from all mothers who participated in this study.

**Culturing NE4C cells and knockdown of MetRS in HEK293T cells**. NE4C cells from ATCC (ATCC number: SCRC-CRL-2925™) were cultured in Eagle MEM and seeded in a plate that had been precoated with poly-L-lysine. All media were supplemented with 10% fetal bovine serum (FBS), 100 × Glutamax, 100 × non-essential amino acids, and 100 × penicillin–streptomycin, purchased from Thermo Fisher.

To investigate the source of protein homocysteine modifications, NE4C cells were starved using 1% FBS medium for 24 h, after which 0.1, 0.5, and 1 mM DL-homocysteine (Hcy) (H4628, Sigma) or L-HTL hydrochloride (HTL) (H6503, Sigma) were added to the complete medium for 8 h. Cells without Hcy or HTL treatment were used as a control.

MetRS knockdown in HEK293T cells were achieved by shRNA virus infection. Interfering sequence TTAAGAAGCCTCAGTGTAA was cloned into PMKO plasmid and co-transfect it with pVSV-G and pGAG-POL plasmids into HEK293T cells to generate viruses. The viruses were obtained after incubating the transfected cells in puromycin containing medium for 36 h after transfection. The knockdown effects were verified by either RT-PCR or by western blotting. The MetRS knockdown HEK293T cells were cultured in DMEM, supplemented with 10% (vol/vol) FBS and 1 mg/ml puromycin.

**Histone extraction**. Core histone proteins were extracted from the tissues or cells using acid extraction[59]. The samples was first homogenized in lysis buffer (10 ml solution containing 10 mM Tris–Cl pH 8.0, 1 mM KCl, 1.5 mM MgCl$_2$, and 1 mM dithiothreitol (DTT)) and chilled on ice. Protease and phosphatase inhibitors were added immediately before lysis of cells, and nuclei were isolated by centrifugation (1500$g$ for 10 min). For the preparation of histones, nuclei were incubated with four volumes of 0.2 N sulfuric acid for overnight at 4 °C. The supernatant was precipitated with 33% trichloroacetic acid and followed by centrifugation (12,000$g$ for 20 min). The obtained pellet was washed with cold acetone and subsequently dissolved in distilled water. The samples were stored at −80 °C before analysis (also showed in our previous papers)[26].

**Fig. 7** Histone H3K79Hcy regulates the expressions of NTDs related genes. **a** ChIP-seq (upper) and RNA-seq (bottom) analysis on *Smarca4*, *Cecr2*, and *Dnmt3b* genes in control and HTL-treated NE4C cells. **b** ChIP-qPCR analysis of histone H3K79Hcy enrichment on *Smarca4*, *Cecr2*, and *Dnmt3b* genes in control and HTL treated NE4C. Data represent mean ± SEM ($n = 3$). *$p$<0.05, **$p$<0.01, ***$p$<0.001 vs. control NE4C groups; the $p$ values were calculated with unpaired $t$ test. **c** ChIP-qPCR analysis of histone H3K79Hcy enrichment in different regions of *Smarca4* in NE4C and HTL treated NE4C cells. Region 1 is at 8000 bp upstream of the *Smarca4* gene; Region 2 is the gene body of *Smarca4*; Region 3 is at 8000 bp downstream of the *Smarca4* gene. Data are represented as means ± SEM ($n = 3$); *$p$ < 0.05 vs. with controls, the $p$ values were calculated with unpaired $t$ test. **d** RT-PCR analysis on gene expression of *Smarca4*, *Cecr2*, and *Dnmt3b* in control and HTL treated NE4C. Data represent mean ± SEM ($n = 3$). **$p$<0.01, ****$p$<0.0001 vs. control NE4C groups; the $p$ values were calculated with unpaired $t$ test. **e** Aberrant expression of histone H3K79Hcy in high-Hcy NTDs. Left panel: Hcy levels in 10 normals and 10 NTDs from human fetal brain tissues; Column normals: 3.322 ± 0.4988 (pmol/mg), $n = 10$; Column NTDs: 41.00 ± 3.705 (pmol/mg), $n = 10$; ****$p$ < 0.0001 vs. normal brain tissues. Middle panel: Western blotting of H3K79Hcy modification from normal and NTDs human fetal brain tissues; Right panel: Quantification of the western blotting signal intensity shown in scatter plot. Column normals: 0.3033 ± 0.02914, $n = 10$; Column NTDs: 0.4368 ± 0.04549, $n = 10$; *$p$ = 0.0237, NTDs vs. normal brains. The $p$ values were calculated with unpaired $t$ test. **f** Nanostring analysis of mRNA level of *SMARCA4*, *CECR2*, and *DNMT3B* in 10 normals and 10 NTDs human fetal brain tissues. *SMARCA4* normals: 3943 ± 345.5, *SMARCA4* NTDs: 2577 ± 158.6; $P = 0.0021$; *CECR2* normals: 724.5 ± 60.35, *CECR2* NTDs: 501.2 ± 47.12, $P = 0.0092$; *DNMT3B* normals: 191.2 ± 22.71, *DNMT3B* NTDs:106.5 ± 19.66, $P = 0.0114$. *$p$<0.05, **$p$<0.01, NTDs vs. normal brains. The $p$ values were calculated with unpaired $t$ test

**Generation of the pan- anti- KHcy antibody**. The anti-KHcy antibodies were developed according to a method previously described for the generation of the anti-KAc antibodies[60]. First 1 mg/ml Bovine Serum Album (BSA) was homo-cysteinylated by incubating with 1 mM HTL under room temperature for 14 h. The KHcy modified BSA was purified by passing reaction mixtures through a Sephadex G-25 gel filtration column in 50 mM Tris buffer in an AKTA-FPLC system to remove organic reagents. Then the proteins were diluted in saline 0.9% (wt/vol) sodium chloride to a final concentration of 0.5 mg/ml and were used to immunize rabbits. The antiserum were collected after four rounds of immunization and the antibodies were affinity purified using affinity purification column with cross-linked synthesized Hcy-lysine containing peptides. The reactivity and specificity of the antibodies were confirmed through a K-Hcy antigen competition experiment (Fig. 1c) and elimination experiments (Fig. 1d). The competition experiment was carried out by incubating the anti-KHcy antibodies with K-Hcy-lysine, while the elimination experiment was performed by incubating a membrane containing unmodified OVA, Suc-OVA, Ac-OVA, and K-Hcy-OVA with the anti-K-Hcy antibody.

**HTL treatment in chicken embryos**. HTL was diluted in Tyrode's buffer (1 mM glucose, 3.5 mM potassium chloride, 100 mM sodium chloride, 0.02 mM phosphate monosodium, 12 mM sodium bicarbonate). Phenol red was added to visualize the embryo and confirm that it was at the appropriate embryonic stage of development. The fertilized chicken eggs (White Leghorns, received from China agricultural university laboratory) were incubated in a humidified incubator at 37 °C for 28–30 h. A total of 0.5 μl of diluted HTL buffer was micro-injected into the neural tube groove using a glass micropipette under a dissecting microscope. The eggs were sealed and incubated for another a series of days, to allow for complete development of the nervous system prior to capturing images. The control group eggs were injected with the same volume of Tyrode's buffer and phenol red. Animal welfare and experimental procedures conformed to the Institutional Guidelines of the Care and Use of Laboratory Animals at China Agricultural University (Beijing, China). All the animal experiments were approved by the Animal Ethics Committee of the Capital Institute of Pediatrics.

**MetRS overexpression plasmid generation and transfection**. Human MetRS cDNA (NM_004990.3) was cloned in pRK7-Flag vector to generate MetRS expression plasmid. HEK293T and NE4C cells were used for transfecting the MetRS plasmid (received from Fudan University). A total of 12 μl of Lipofecta-mine[2000] transfection reagent (11668-019, Thermo Fisher) diluted in 125 μl of Opti-MEM (51985-034, Thermo Fisher) was prepared. In addition, 4 μg of MetRS plasmid and 4 μg of empty vector were diluted with 125 μl of Opti-MEM and incubated for 5 min at room temperature. The 250 μl mixture of Lipo2000 and plasmids was added into one well. After 5–6 h, the medium was renewed and the cells were incubated for 24–96 h for further use in the following experiments.

**Western blotting**. Histone mixture (5 μg) was separated on a NuPAGE™ 12% Bis-Tris Gels, then transferred electrophoretically onto a Hybridization Nitrocellulose Filter. The membrane was prehybridized in Tris-buffered saline (TBS)(0.9% NaCl, 10 mM Tris–HCl, pH 7.5) containing 0.05% Tween 20 (TBST) and incubated for 1 h at room temperature in TBST containing 10% nonfat skimmed milk. Then, it was transferred to a solution containing 5% milk/TBST and primary antibody and incubated overnight at 4°C. After washing with TBST buffer, the membrane was immersed in 5% milk/TBST containing horseradish peroxidase (HRP)-conjugated secondary antibody (Cat# SC-2048, Zhongshan Jinqiao) for 1 h. The membrane was washed with TBST buffer, developed using the ECL system, and exposed to X-ray film.

The following primary antibodies were used: H3K4me3 (39159, Active Motif), H3K9me3 (39285, Active Motif), H3K27me3 (61017, Active Motif), H3K79me2 (39923, Active Motif), methylated (εN) lysine antibody (ICP0501, ImmuneChem), acetyl lysine antibody (ICP0380, ImmuneChem), Hcy antibody (obtained from Fudan University, Shanghai, China), and H3K79Hcy antibody (Jing Tiancheng Company, Beijing, China). The following secondary antibodies purchased from Zhongshan Jinqiao Biological Company were used: rabbit IgG(H + L)/HRP (ZB-2301, ZSGB-Bio) and mouse IgG(H + L)/HRP (ZB-5305, ZSGB-Bio).

All uncropped western blots can be found in Supplementary Figures 5–12.

**ChIP- seq**. A SimpleChIP® Enzymatic Chromatin IP Kit (Cell Signaling Technology, Massachusetts, USA) was used for the ChIP assays, in accordance with the manufacturer's protocol. Formaldehyde cross-linked chromatin was obtained from about $8 \times 10^7$ NE4C cells. Cross-linked chromatin was immunoprecipitated with antibodies to H3K4me3, Hcy and H3K79Hcy overnight at 4°C. Normal rabbit IgG were used as negative control. Immunoprecipitated DNA was analyzed by sequencing. Indepth whole-genome DNA sequencing was performed by BGI (www.genomics.org.cn, BGI, Shenzhen, China). The raw sequencing image data were examined by the Illumina analysis pipeline, aligned to the *Mus musculus* reference genome (UCSC, mm9) using Bowtie 2, and further analyzed by MACS (Model-based Analysis for ChIP-Seq; https://github.com/taoliu/MACS). Enriched binding peaks were generated after filtering through control input. Raw data files

for ChIP sequencing have been deposited in the NCBI Gene Expression Omnibus database under the accession codes GSE104093.

**RNA- seq and analysis**. RNA samples were collected from normal cultured NE4C and HTL treated NE4C. Library construction and sequencing were performed on a BGISEQ-500 by Beijing Genomic Institution (www.genomics.org.cn, BGI, Shenzhen, China). Clean-tags were mapped to the reference genome and genes available at the Mice Genome. For gene expression analysis, the matched reads were calculated and then normalized to RPKM using RESM software[61]. The significance of the differential expression of genes was defined by the bioinformatics service of BGI according to the combination of the absolute value of log2 Ratio ≥ 1 and FDR ≤ 0.001. Raw data files for RNA sequencing have been deposited in the NCBI Gene Expression Omnibus database under the accession codes GSE104094.

**RNA extraction and nanostring for mRNA detection**. Total RNA was extracted from the brain tissues of humans using the RNeasy Mini kit according to the manufacturer's instructions (Qiagen, Mississauga, Canada). The NanoString nCounter detection method was used to examine genes' transcript levels in human tissues. Hybridizations were performed according to the NanoString Gene Expression Assay manual. Approximately 100 ng of each RNA sample was mixed with 20 μL of nCounter Reporter probes and 5 μL of nCounter Capture probes in hybridization buffer. The purified target/probe complexes were eluted and immobilized in a cartridge for data collection, which was conducted in the nCounter Digital Analyzer. The results were normalized to the *GAPDH*, *CLTC* and *GUSB* genes.

**RT-PCR**. According to the results of sequencing, several pairs of primers were designed (Supplementary Table 4) for the primers used in this experiment. RT-PCR was performed using RT2 SYBR® Green ROX™ qPCR Mastermix (Qiagen, Mississauga, Canada), in accordance with the manufacturer's instructions. Amplification, data acquisition and analysis were carried out using QuantStudio 7 Flex (Applied Biosystems, Singapore). The percentage of DNA brought down by ChIP was calculated using the $2 - \Delta CT$ method. Three independent ChIP experiments were performed for each analysis.

**HPLC/MS**. The tryptic digests were injected into an UltiMate 3000 RSLCnano System (Dionex, Germering, Germany) and analyzed by a Q Exactive HF mass spectrometer (Thermo Scientific, Bremen, Germany). Full-scan MS spectra in the $m/z$ range of 350–2000 were acquired in the Orbitrap. Twenty of the most intense ions were isolated for MS/MS analysis. The raw data were processed using Proteome Discoverer (version 2.1.0.81, Thermo Fisher Scientific), searching with a database of human histone (www.uniprot.org, accessed October 2015). Peptides were generated from a semi-tryptic digestion with up to four missed cleavages, carbamidomethylation of cysteines as a fixed modification, and oxidation of methionines as a variable modification. Precursor mass tolerance was 20 ppm and product ions were searched at 0.05 Da tolerance. Peptide spectral matches (PSMs) were validated using a percolator based on $q$-values at a 1% false discovery rate (FDR). The modified peptides passing the FDR were exported to a text file and processed by PRM. The area of peaks was used to represent the number of modifications.

**PRM**. Raw data were searched against the corresponding histone database. The modification include lysine homocysteinylation, acetylation, and mono-, di- and trimethylation were searched. The mass inclusion list involved mass, charge, polarity and the time from start and end. The full scan method was as described above. The PRM method employed an Orbitrap resolution of 30,000 (at $m/z$ 350) and a target AGC value of 2e5. The precursor ions of each peptide were duplexed using ± 0.8 $m/z$ unit windows. Each sample was analyzed in triplicate.

**PRM data analysis**. PRM data were manually curated within the Xcalibur Qual Browser (version 4.0.27.19; Thermo Fisher Scientific) and through the use of Skyline (version 3.5.0.9319; AB Sciex). In Xcalibur Qual Browser, the determination of the area under the curve (AUC) of selected fragment ions was based on the presence of product ion signals within ± 2.5 min of the expected retention time, with mass error within ± 5 ppm. Skyline used raw files as input to generate and extract modified peptide normalized area at a 0.05 $m/z$ ion match tolerance for each PRM spectrum. The skyline detected results were further confirmed by area calculation of the raw data as shown in Supplementary Figure 2G.

**HTL treatment in vitro**. Purified histones, including H2a (M2502S, NEB), H2b (M2505S, NEB), H3 (M2503S, NEB) and H4 (M2504S, NEB) were selected. Reactions were carried out at 37 °C in mixtures containing 0.5 mM L-HTL hydrochloride, 10 μg/50 μl histones, 0.1 M sodium phosphate buffer (pH 7.4), 0.2 mM EDTA. A series of concentration (from 0 to 10 mM) and time gradients (from 0 to 14 h) were used for following experiments.

**Generation of anti H3K79Hcy antibody**. Anti-H3K79Hcy antibody was generated and purified from rabbit with lysine-homocysteine modified bovine serum albumin (BSA) as an antigen. To generate H3K79 site-specific antibody, the synthesized peptide CREIAQDFK(Hcy)TDL was used as an antigen for rabbit immunization. Antiserum was collected after four sessions of immunization. The antibody was done by AbMax Biotechnology Co., Ltd. To test the specificity of the anti-H3K79Hcy antibody, three experiments were designed as Supplementary Figure 2A, 2B and 2C. The dot-blot results showed that H3K79Hcy antibody could strongly recognize the homocysteinylated H3K79 peptide, weakly recognized the H3K79 peptide, but almost not recognized the dimethylated H3K79 peptide, H3K27 peptide, homocysteinylated H3K27 peptide, H3K115 peptide, and homocysteinylated H3K115 peptide (Supplementary Figure 2A). Two additional experiments were conducted to verify the specificity of the anti-H3K79Hcy antibody. A significantly stronger signal was detected with anti-H3K79Hcy antibody on homocysteinylated H3, compared to that of unmodified H3 (Supplementary Figure 2B). However, such a strong reactivity to homocysteinylated H3 can be effectively blocked by pre-incubation with increasing amount of H3K79Hcy peptide, while baseline reactivity to unmodified H3 remains unchanged (Supplementary Figure 2B), confirming the specificity of the anti-H3K79Hcy antibody. Supplementary Figure 2C shows that increasing levels of histone homocysteinylation in NE4C cells was detected using anti-H3K79Hcy antibody with increasing concentration of HTL. These three experiments support the validation that anti-H3K79Hcy antibody specifically recognizes homocysteinylated H3K79.

**Hcy level detection in brain tissue**. Hcy level detection is set up by our laboratory[62]. Brain tissue were treated with 150 mL of 50 mM DTT and waited a 20 min period in room temperature to reduce disulfide bonds, then 200 uL of internal standard (Hcy-d4) were added. Spiked brain samples were vortexed and homogenized before a 15 min sonication and a 12,000g centrifuge. The supernatant were transferred to a solid phase extraction (SPE) tip in a commercial kit named EZ:faast (KH0-7337, Phenomenex), sample purification and Hcy derivatization was conducted according to the manufacturer's protocol. After that, the derivative Hcy was evaporated and re-dissolved using methanol–water (65: 35, v/v) containing 1 mM ammonium formate before injection. An Agilent 6410B triple-quadrupole mass spectrometer with an Agilent 1200 system HPLC (Palo Alto, CA, USA) were used for LC-MS/MS analysis. Separation was performed on a Zorbax Bonus-RP column (100 mm*2.1 mm i.d., 1.8 mm particle size, Agilent Technologies, Germany) at a flow rate of 0.25 mL/ min. The mobile phase was methanol–water (65: 35, v/v) containing 1 mM ammonium formate. Each sample was injected in a volume of 1 mL via an auto-sampler and separated by isocratic elution in 6.5 min. The column temperature was 35 °C. The MS/MS experiments were performed under positive-ion (ESI + ) mode with multiple-reaction monitoring (MRM). The capillary voltage was set to 4 kV and the source temperature was set to 350 °C. Nitrogen served as the nebulizer gas at a flow rate of 10 L min/1 and a pressure of 45 psi. High purity nitrogen was used as the collision gas. The MRM transition for Hcy and Hcy-d4 were 350-204.1 and 354.2-208.1, respectively.

**Statistical analysis**. Statistical parameters for each experiment are reported in the corresponding figures. All data presented were derived from three independent experiments and were reported as standard error of the mean (SEM).

**Data availability**. We declare that all data supporting the findings of this study are available within the article and its supplementary information files or from the corresponding author upon reasonable request. Raw data files for ChIP sequencing have been deposited in the NCBI Gene Expression Omnibus database under the accession code GSE104093. Raw data files for RNA sequencing have been deposited in the NCBI Gene Expression Omnibus database under the accession code GSE104094.

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

## Acknowledgements

This study is supported by the National "973" project (2013CB945404), CAMS Initiative for Innovative Medicine (2016-I2M-1-008), Beijing municipal program of medical research (Grant No. 2016-04), the National Natural Science Foundation of China, Beijing, China (81741044 and 81771584), and Beijing Natural Science Foundation (7182024)

## Author contributions

T.Z. initiated and conceived the study; T.Z., S.M.Z., B.L.B., X.Y.M., and Q.Z. designed and performed the experiments; T.Z., Z.Y.Q., B.L.B., X.Y.M., and Q.Z. wrote the manuscript; Q.Z. verified the Hcy modification; X.Y.M. verified the Hcy modification; B. L.B. characterized the functions of Hcy in cells; C.L.W. and J.Y.Z. characterized the functions of Hcy in mouse; H.Y.C and D.L. did the bioinformatics and biostatistics analysis under the guidance of J.X.W., Z.G.W. and H.Y.W.; B.L.B. carried out in vitro transcription assay under the guidance of S.M.Z.; C.L.W., D.L., G.Q.D., J.S.H., and Q.Z. contributed to mass spectrometric analysis; C.L.W., D.L., Q.Z., and H.Y.C. contributed to animal models and animal analysis; M.Z. helped with Hcy level testing in human brain; S.W. helped with the nanostring experiment for detection of mRNA in human brain; L. W., Q.X., J.X.W., Z.Y.Q., S.M.Z., H.Y.W., J.S.H., and G.Q.D. provided important feedback to the manuscript.
