## [Peer Review File · Nature Communications]

Reviewers' Comments:

Reviewer #1:

Remarks to the Author:

Zhang et al. report that homocysteine (Hcy) levels are elevated in brains of human fetuses with neural tube defects relative to normal fetus brains and examine downstream consequences of elevated Hcy. They identify numerous N-homocysteinylation sites in brain histones, show that homocysteinylation of Lys79 in histone 3 (H3K79Hcy) is associated with reduced expression of neural tube closure genes (*Smarca4*, *Cecr2*, *Dnmt3b*), and conclude that these processes are involved in the etiology of neural tube defects. These are important novel findings that would of interest to others in the community and in the wider field. The work is highly likely to influence the thinking in the field and will open up new avenues of research. Although the work appears to be convincing, it would be strengthened by satisfactorily addressing the following issues.

Specific comments:

1. In their HPLC/MS/MS analyses Zhang et al. use a mass increase of 171.0376 Da (monoisotopic mass) for the identification of lysine N-homocysteinylation (N-Hcy-Lys) in specific histone peptides. As no experimental detail is provided, it is not clear what exactly this value reflects. Further, their statement on lines 77-78 that "Such as mass shift resembled what has been previously published..." is incorrect. In fact, the mass increase of 171.0376 Da does not resemble and is quite different from the value of 174.0 Da for a mass increase due to N-homocysteinylation in the original reports by other investigators, e.g., Glowacki & Jakubowski, *JBC* 2004; Perla-Kajan et al., *Biochemistry* 2007; Marczak et al., *J Proteomics* 2011; Sikora et al., *Amino Acids* 2014. As using the incorrect 171.0 Da value would not lead to correct identification of N-Hcy-Lys sites, one has to assume that the 171.0 value is due to a typographical error. Please clarify/correct.
2. Lines 83-84, Table 1: Several of N-Hcy-peptides identified by Zhang et al. have structures incompatible with the specificity of trypsin, which does cut after modified Lys residues. Thus tryptic peptides cannot have N-Hcy-Lys at the C-terminus. However, according to Zhang et al. H2A peptide with the modification at K15, H2B peptide with the modification at K125, and H4 peptides with the modifications at K16 and K79, all have the modification as their C-termini. As these findings are not consistent with previous studies showing that the N-Hcy-Lys modification is present only at an internal lysine residue in tryptic peptides, an explanation is required.
3. Lines 89-90: The authors state that they confirmed the presence of KHcy in histones from human embryonic tissues and in other species by Western blotting using "an antibody against lysine homocysteinylation (anti-KHcy)". Unfortunately, no rigorous data are provided regarding the antigen specificity of their anti-KHcy antibody.
4. Lines 137-138: The statement "Inside cells, Hcy can be converted to HTL under catalysis of the cellular enzyme MetRS" should be supported by source citation.
5. Line 146, Fig.S1B: Info regarding MetRS overexpression and MetRS knockout is missing. The MetRS experiments are missing important controls: quantification of MetRS expression to demonstrate that its levels have changed as intended. The presence of H3K790Hcy signal in panel E suggests that the expression of MetRS is not fully blocked in the KO cells or that MetRS is not the only source of HTL.
6. Fig. 3 and 4: Describe what each of the 3 Ponceau-stained bands represents.
7. Lines 170-171, Fig. S3A: Zhang et al. also use another antibody, anti-H3K79-Hcy, and state that "This antibody was found to specifically bind to the modified H3K79Hcy peptide (Fig. S2A)." However, Fig. S2A clearly shows that the antibody binds as well to unmodified H3K79 peptide and, less efficiently, to two other H3 peptides. Again, the manuscript would be strengthened by providing a more convincing and quantitative information regarding the antigen specificity of the anti-H3K79-Hcy antibody.
8. Line 210-214: In reference to NTD experiments using a chick embryo model (Fig. 5), the authors state that "Higher levels of histone Hcy and H3K79Hcy were detected in samples from chicken of HTL-treated group with phenotypes of NTDs (Fig. 5D)." Showing single band cut-outs from two samples is not convincing to support this statement. A more rigorous quantitative analysis would be required in order to determine whether the increase is statistically significant.
9. Lines 248-249: The authors state that H3K79Hcy-containing nucleosomes reduce NTC gene

expression but upregulate the global gene expression. They also state that HTL-treatment of NE4C cells, which increases levels of H3K79Hcy, decreases occupancy of NTD-related genes by H3K79Hcy-containing nucleosomes (lines 264-269). These statements appear to be contradictory and require clarification.

10. Lines 259-274: The findings that H3K79Hcy enrichment in NTC-related genes is reduced in response to the treatment with HTL (Fig. 7A-C) appears to be inconsistent with the finding that the treatments with HTL increase levels of N-Hcy-histone (Fig. 3C), including H3K79Hcy (Fig. S2B). The explanation of this dissonance provided in lines 405-407 of the Discussion that "higher level of H3K79Hcy has a negative impact in its binding to aforementioned genes" is not convincing.

11. Lines 287-280, Fig. 7E, Table S1: The listed Hcy values are several orders of magnitude higher than published tissue Hcy values, which are about 50-100 pmol/mg in normal tissues and about 5000 pmol/mg in severe HHcy. Please check your calculations.

12. Line 335-336: the list of N-Hcy-proteins has grown substantially since 1999. See Jakubowski H. Homocysteine in Protein Structure Function and Human Disease, Springer, 2013 and more recent articles.

13. Line 341: The statement "...this is the first time that homocysteinylation specific to histone has been reported" is not entirely true. In fact, histone N-homocysteinylation has been examined at least in two labs: e.g., Gurda D et al., Histones are targeted for N-homocysteinylation in human endothelial cells. *Acta biochimica Polonica* 61 Suppl 1: 127, 2014; Xu L et al., Crosstalk of homocysteinylation, methylation and acetylation on histone H3. *Analyst* 140: 3057-3063, 2015.

14. Methods: Description of many procedures is missing, e.g. histone extraction from tissues/cells. Other procedures are missing important details. For example, antibodies that have been generated for this study require a more detailed description/characterization. What is the nature of Mars-KO in HEK293T cells? Is it conditional? How do you grow/treat these cells for experiments? Without these details, the work would be difficult to reproduce.

15. If the N-Hcy-histone modification is as abundant as Zhang et al. report, it should be easily identified and quantified by another method, e.g., a direct chemical assay for Hcy in brain histones, which would greatly strengthen the authors' conclusions.

Reviewer #2:

Remarks to the Author:

This is a very interesting paper that studies homocysteinylation on chromatin. Elevated homocysteine is a hallmark of folate, and B12 deficiencies and thus Neural Tube Defects. The authors identify elevated H3K79Hcy and other homocysteinylation in this setting and begin to study their function. This is an important study because it defines a novel mark on histones and begins to characterize the function in a disease setting which includes the development of an antibody to probe the mark.

Some concerns should be addressed.

- In Figure 1, they used 10 human embryonic brain samples and ID'd 39 unique KHcy sites across the 4 histone variants. Were the majority of these overlapping between the samples? Essentially, how conserved are these 39 marks between different embryos? Also, any ideas about why there would be such tissue specificity (i.e. highly abundant in lung tissue but virtually absent in muscle)?
- Assuming adult human brain samples were used in Fig. 1D, how would the authors predict H3K79Hcy expression patterns to differ between adult vs embryonic samples? Is this mark conserved into adulthood, and does the localization change?
- The authors looked at differences in localization between H3K4me3 (mostly localized to gene promoters) and H3K79Hcy (mostly localized to gene bodies), but does this mark co-localize with other activational epigenetic marks that typically localize in the gene body? How does the

localization of typically repressive marks (like H3K27me3) compare to the localization of H3K79Hcy in these neural development genes?

- In Fig 4B, the authors claim that increases in H3K79Hcy don't affect methylation or acetylation, but it looks to me like acetylation does go down. Maybe this isn't important. Also, in Fig 4G, the authors say that methylation of H3K79 did not significantly change, but the panel shows a significant difference in H3K79me1 (and they didn't look at H3K79me3 or H3K79ac).
- The authors focus the majority of the paper on H3K79Hcy, which they provide sufficient rationale for. However, previous work by the authors showed that aberrant H3K79me2 results in NTD, and the authors don't show any link between these two marks in this paper. Which do they think plays a larger role?
- If Hcy supplementation results in an overall increase in histone homocysteinylation but this mark is significantly reduced in neural development genes, where do they think this increase is occurring? Do they think it's localizing to other genes?
- Somewhat minor, but if MetRS is the enzyme responsible for converting homocysteine to HTL, why would MetRS knockout blunt the effects of HTL supplementation on histone homocysteinylation? (Fig S1B)

Reviewer #3:

Remarks to the Author:

This is an interesting paper that suggests a molecular mechanism for the well know observation that high levels of homocysteine (Hcy) in maternal blood is a risk factor for neural tube defects in the offspring. The manuscript goes from general analysis of homocysteinylation of histones, to showing a connection between Hcy injection and NTDs in an embryonic chick model. Finally human fetal brain samples with high levels of Hcy are shown to have increased levels of H3K79Hcy and on average have decreased expression of 3 NTD genes.

On the whole this study is well done and represents important information. However, it is still correlative to human NTD causation and looks at Hcy levels at a time long after neurulation has occurred. It will be interesting to use the chick model to look at the effect of homocysteinylation at the time of neurulation.

Additionally, there are places where important detail is lacking, and others where I have questions about some interpretations of Figures.

There are several places where more rational or information needs to be give:

- Page 6, Fig 3 – Why were additional experiments only done with H3, when all 4 histones had Hcy sites mapped?

- Why was H3K79Hcy chosen for further study, over the other 5 sites that also were homocysteinylated in untreated NE4C cells and fetal brain samples, although in different histones. (figure 4)

- There are over 300 genes connected with NTDs in the mouse. Why were Smarca4, Cecr2, and Dnmt3b the genes chosen for further analysis? These epigenetic regulator genes were not in the ChIP-Seq list in Table S4 (without a title is it impossible to fully interpret this Table), while epigenetic regulator NTD genes Hdac4, Sirt1 and Smarcc1 are in this list and these were not followed up..

- Methods: Please give more details on the human embryonic brain samples in Table S1. The

references given are not informative enough. How were the controls matched? Was the level of folic acid supplementation in the Moms the same in the controls vs the cases of NTDs? Should an anencephaly be included, since 10/11 cases are spinal defects?

- Also, were the 11 NTD brain samples with high Hcy levels selected from a larger group of NTD samples based on their Hcy levels? If so, how many samples were analysed to give this selected group?

- Figure 1-3 rely on an Anti-Hcy antibody to explore homocysteinylation. The only information on the antibody is that it came from Fudan University (no investigator name given). How was it confirmed that this antibody is specific? Is there a publication? The authors clearly show that their own Anti-H3K79Hcy is specific, but the anti-Hcy is a mystery.

Interpretation of results:

- Fig 2, panel 2, showing HTL treatment of H3 is time dependent. I see no obvious difference between 2 and 6 hours. There is certainly an obvious difference between 2/6 and 14 hours. Make your description more specific – could 14 hrs be starting to show artifacts? Is there any quantitation for these Westerns?

- Line 181-182, Fig 4D – “Furthermore, treatment with HTL in cultured NE4C cells had no obvious effects on histone methylation and acetylation (Fig. 4D).” Looking at this Figure I would say that methylation goes up a little at 0.5mM and then back down again at 1 mM. Also, acetylation is considerably decreased at 1 mM. The above sentence needs to be modified. What would the significance of these differences be?

- Line 186-188, Figure 4 G & H. “Compared to H3K79Hcy, level of other histone modifications, i.e. methylation on H3K79 did not change significantly following HTL treatment (Fig. 4G and 4H).” But Figure 4G shows a significant difference between control and HTL treatment!

- In Figure 7, I find panels A and B difficult to reconcile with panels C and D. In A (ChIP-Seq) only Ccnc2 shows a large difference, but in C (ChIP-PCR) Dnmt3b shows the biggest difference. Panel B doesn't match panel d. I understand these panels compare different techniques, but some explanation should be given. Also, why is Smarca4 the only gene analysed in more depth (panel B)?

Other comments:

- Please put Titles and brief explanations on all Supplementary Tables! None currently have titles.

- Introduction: What is the connection between Hcy levels and folate supplementation? If folate is supplemented, does the connection between Hcy and NTDs disappear?

- There was no mention of homocysteine in mouse models of NTDs. Bennett et al (2006) found that an increase in maternal homocysteine did not result in neural tube defects. Greene et al (2003) cultured mouse embryos in 0.5mM and higher homocysteine and saw signs of toxicity but no NTDs, concluding that high levels of homocysteine may represent a metabolic disturbance rather than something causal. Since the authors are using mouse models to identify NTD genes, there should be some comment on whether the mouse can be used as a model to study the role of homocysteine levels in NTDs. Could any of their findings be tested in mice? Or is this more evidence for the utility of the chick model?

- Figure 5, chick experiment. Is 0.5 mM HTL close to the levels that human embryos might experience, or is it much larger? Did you try lower doses and still see NTDs?

- Line 74 – clarify which 10 brain samples were used (I presume the 10 normal samples in Table S1).

- Line 125-127 – “It is worth mentioning that the number of homocysteinylation sites seemed to correlate with the intensity of KHcy signal on western blotting (Figure 2B top panel and 2E).” H4 does not fit the pattern.
- Figure S2 – There is no description of panel C
- Line 197 – Why is “similar NTC gene distribution on chromosomes as human embryos” an advantage for the chick system?
- Line 207: “When the same amount of HTL was injected into placenta, similar results were obtained with regarding rates of embryo survival rate, overall malformations, and occurrence of NTDs (data not shown).” Please add the numbers or remove. Please clarify – what do you mean by injecting into the chicken “placenta”.
- Is Table S4 a list of CHIP-Seq-detected genes from using anti-H3K4me3 or anti-H3K79KHcy or both?
- Lines 356-359 – This is a very long and convoluted sentence that is very difficult to understand.
- Lines 387-391 – this passage in the Discussion gives new results that were not discussed in the results section, and refers to Fig. 4S. Furthermore, the list of genes given here are the ones they tested, but Fig 4S shows that not all of them showed significant decrease in expression. This section should be in the results and identify only the genes that showed an increase. Then it can be discussed in the discussion.
- Line 395 – Do the authors mean “binding of H3K79-Hcy ..?”
- Line 707 – name the unrelated control antibodies used
- Table 7 headings are run together and not readable. Also, the Table would be much more useful if gene name symbols were also included along with gene ID numbers. The lines are fragmented into 3 non-consecutive pages, making it very difficult to look at anything specific.
- The difference in Hcy levels in human embryo brains is very impressive in Figure 7E first panel. This leads to a wide variation in Figure 7E 3rd panel where H3k79Hcy modification is measured, with many individuals within the normal range. This could be due to many factors, considering that the Hcy levels are being measured long after neurulation has occurred, different genetic backgrounds may show different resistance to high Hcy levels, and there may be many different causes for the 11 NTDs. I wonder if the subset of individuals above the normal range of H3K79Hcy were analysed separately for panel F, if you would find a bigger difference in expression of the 3 NTD genes.

Reviewers' comments:

Reviewer #1 (Remarks to the Author):

Zhang et al. report that homocysteine (Hcy) levels are elevated in brains of human fetuses with neural tube defects relative to normal fetus brains and examine downstream consequences of elevated Hcy. They identify numerous N-homocysteinylation sites in brain histones, show that homocysteinylation of Lys79 in histone 3 (H3K79Hcy) is associated with reduced expression of neural tube closure genes (*Smarca4*, *Cecr2*, *Dnmt3b*), and conclude that these processes are involved in the etiology of neural tube defects. These are important novel findings that would of interest to others in the community and in the wider field. The work is highly likely to influence the thinking in the field and will open up new avenues of research. Although the work appears to be convincing, it would be strengthened by satisfactorily addressing the following issues.

Response:

First, we wish to express our sincere thanks for the reviewer's positive and encouraging comments. The reviewer's insightful questions and constructive suggestions have been thoroughly discussed among all authors and we have made every effort to address the reviewer's questions in the following.

Specific comments:

1. In their HPLC/MS/MS analyses Zhang et al. use a mass increase of 171.0376 Da (monoisotopic mass) for the identification of lysine N-homocysteinylation (N-Hcy-Lys) in specific histone peptides. As no experimental detail is provided, it is not clear what exactly this value reflects. Further, their statement on lines 77-78 that "Such as mass shift resembled what has been previously published..." is incorrect. In fact, the mass increase of 171.0376 Da does not resemble and is quite different from the value of 174.0 Da for a mass increase due to N-homocysteinylation in the original reports by other investigators, e.g., Glowacki & Jakubowski, JBC 2004; Perla-Kajan et al., Biochemistry 2007; Marczak et al., J Proteomics 2011; Sikora et al., Amino Acids 2014. As using the incorrect 171.0 Da value would not lead to correct identification of N-Hcy-Lys sites, one has to assume that the 171.0 value is due to a typographical error. Please clarify/correct.

Response:

We are very apologetic about the typographical error regarding the mass shift of N-homocysteinylation in the manuscript. In fact, we did use a mass increase of 174.04600Da (monoisotopic mass) for the identification of lysine N-homocysteinylation (N-Hcy-Lys) in specific histone peptides. In the revised manuscript, we incorporated the correct value of 174.04600 for mass increase derived from N-homocysteinylation at lysine residues of the digested histone peptides. We wish to thank the reviewer for pointing out the error.

2. Lines 83-84, Table 1: Several of N-Hcy-peptides identified by Zhang et al. have structures

incompatible with the specificity of trypsin, which does cut after modified Lys residues. Thus tryptic peptides cannot have N-Hcy-Lys at the C-terminus. However, according to Zhang et al. H2A peptide with the modification at K15, H2B peptide with the modification at K125, and H4 peptides with the modifications at K16 and K79, all have the modification as their C-termini. As these findings are not consistent with previous studies showing that the N-Hcy-Lys modification is present only at an internal lysine residue in tryptic peptides, an explanation is required.

Response:

We thank the reviewer for raising the issue regarding the substrate specificity of trypsin. Indeed, it has been generally accepted that derivatized lysine and arginine residues are not optimal substrates during a trypsin digestion and such substrate specificity works to a good advantage in identifying and localizing modifications on lysine and arginine in histones and other proteins¹⁻³. However, there has been emerging evidence demonstrating cleavage by trypsin leading to the generation of C-terminus modified residues of arginine and lysine.

Fenselau's group reported cleavage by trypsin to produce a glycinyglyciny-lysine residue at C-terminus after protein ubiquitination^{4,5}. In addition, others have shown that trypsin could digest some modified lysine such as propionylation on H3K4^{6,7}. In our study, six different homocysteiny-ated peptides were detected using mass spectrometry, including KGLGKGGAK_{Hcy}RHR, KGGAK_{Hcy}R, GAK_{Hcy}RHRKVLRL, GKGGKGLGKGGAK_{Hcy}R, GGKGLGKGGAK_{Hcy}RHR, and GRGKGGKGLGKGGAK_{Hcy}, with regarding to the H4K16 site. The presence of the first five peptides provided solid evidence that H4K16 could be homocysteiny-ated. A series of b- and y-type homocysteiny-ation fragment ions from the raw MS data supported the notion of an Hcy shift. Therefore, we concluded that the sixth peptide, GRGKGGKGLGKGGAK_{Hcy}, was indeed present and further confirmed that C-terminus K_{Hcy} could be digested by trypsin.

3. Lines 89-90: The authors state that they confirmed the presence of KHcy in histones from human embryonic tissues and in other species by Western blotting using "an antibody against lysine homocysteiny-ation (anti-KHcy)". Unfortunately, no rigorous data are provided regarding the antigen specificity of their anti-KHcy antibody.

Response:

We thank the reviewer for pointing out the lack of validating information for anti-Hcy antibody.

Initially, the information on the generation and validation of the anti-K-Hcy antibody was part of a separate manuscript. Now we decided to incorporate such information in the present manuscript to strengthen our data set. In the revised manuscript, we included two additional figures, 1C and 1D, to demonstrate the specificity of the anti-K-Hcy antibody, as well as the description of results in lines 91-98. Detailed information for the generation of the antibody

was also added to the method section.

4. Lines 137-138: The statement “Inside cells, Hcy can be converted to HTL under catalysis of the cellular enzyme MetRS” should be supported by source citation.

Response:

Thanks for the reviewer’s suggestion. A reference pertaining the enzymatic activity of MetRS titled “ Synthesis of homocysteine thiolactone by methionyl-tRNA synthetase in cultured mammalian cells” was on line 153 in the revised manuscript.

5. Line 146, Fig.S1B: Info regarding MetRS overexpression and MetRS knockout is missing. The MetRS experiments are missing important controls: quantification of MetRS expression to demonstrate that its levels have changed as intended. The presence of H3K790Hcy signal in panel E suggests that the expression of MetRS is not fully blocked in the KO cells or that MetRS is not the only source of HTL.

Response:

We are sorry for the fact that the description for Fig.S1B in the original manuscript is misleading. The MetRS-KO HEK293T cell, as a matter of fact, is a HEK293T cell line with stable MetRS knockdown. Our experimental data revealed that MetRS targeting shRNA were stably produced in this cell line, resulting in an inhibition efficiency of 90% of MetRS expression.

Fig.S1B shows histone homocysteinylation level in MetRS knockdown HEK293 cells. In this particular experiment, we did not treat cells with HTL. Under MetRS knockdown conditions, Hcy would not be fully converted to HTL. Therefore, HTL level would decrease, leading to the decrease in the level of histone homocysteinylation.

6. We have corrected errors and added detailed information to the material section in the revised manuscript. We also added FigureS1C and FigureS1D to show the MetRS level in MetRS overexpression and MetRS knockdown experiment. Fig. 3 and 4: Describe what each of the 3 Ponceau-stained bands represents.

Response:

Thanks for the reviewer’s suggestion. The description of the 3 Ponceau-stained bands was incorporated in Figures 3 and 4.

7. Lines 170-171, Fig. S3A: Zhang et al. also use another antibody, anti-H3K79-Hcy, and state

that “This antibody was found to specifically bind to the modified H3K79Hcy peptide (Fig. S2A).” However, Fig. S2A clearly shows that the antibody binds as well to unmodified H3K79 peptide and, less efficiently, to two other H3 peptides. Again, the manuscript would be strengthened by providing a more convincing and quantitative information regarding the antigen specificity of the anti-H3K79-Hcy antibody.

Response:

We thank the reviewer for the critical data review and constructive suggestion.

We agree with the reviewer’s comment that Fig.S2A in the previously submitted manuscript did not properly address the question on specificity of the H3K79Hcy antibody. Therefore, we conducted a series of experiments as suggested by the reviewer (a representative chart shown below).

The H3K79Hcy antibody generated in our lab is a rabbit polyclonal antibody. Our data indicated that this antibody has an acceptable specificity at lower antigen concentrations. The chart above shows that when 0.25 μ g peptide was used in dot-blot experiments, strong reactivity of anti-H3K79Hcy antibody was detected only to the H3K79Hcy peptide, and to a much lesser degree, to unmodified H3K79 backbone. With an increasing amount of antigens used, the specificity of the antibody decreases gradually in dot-blot experiments. When 1 μ g of peptides were used, significant reactivity of the antibody was seen not only to H3K79Hcy, but to unmodified H3K79 as well, along with a detectable level of reactivity to H3K79me2 peptide. At a 5 μ g peptide load, antibody specificity diminished and the difference in binding to H3K79Hcy, H3K79, and H3K79me2 appeared to be insignificant. In addition, weaker reactivity of the antibody was also observed for H3K115 peptide and homocysteinylated H3K115 peptide. Data from above-described experiments could serve as an explanation for noted non-specific binding of the antibody in Fig. S2A of the previously submitted manuscript, as the peptide load of 2 μ g was used during the experiment conducted prior to the initial submission.

Two additional experiments were conducted to verify the specificity of the anti-H3K79-Hcy antibody, the results are shown in Fig. S2B and Fig. S2C of the revised manuscript. A significantly stronger signal was detected with anti-H3K79Hcy antibody against homocysteinylated H3, compared to that of unmodified H3 (Fig. S2B). However, such a strong reactivity to homocysteinylated H3 can be effectively blocked by pre-incubation with

an increasing amount of H3K79Hcy peptide, while baseline reactivity to unmodified H3 remains unchanged (Fig. S2B), confirming the specificity of the anti-H3K79Hcy antibody. Figure S2C shows that increasing levels of histone homocysteinylation in NE4C cells was detected using anti-H3K79Hcy antibody with increasing concentration of HTL.

Collectively, results from these three experiments presented here (Fig. S2A, Fig.S2B, and Fig. S2C) support the validation that anti-H3K79-Hcy antibody specifically recognizes homocysteinylation H3K79.

8. Line 210-214: In reference to NTD experiments using a chick embryo model (Fig. 5), the authors state that “Higher levels of histone Hcy and H3K79Hcy were detected in samples from chicken of HTL-treated group with phenotypes of NTDs (Fig. 5D).“ Showing single band cut-outs from two samples is not convincing to support this statement. A more rigorous quantitative analysis would be required in order to determine whether the increase is statistically significant.

Response:

We agree with the reviewer’s comment that a rigorous quantitative analysis is needed to support the conclusion that higher levels of histone Hcy and H3K79Hcy are present in samples from chicken of HTL-treated group with phenotypes of NTDs.

We conducted an additional experiment with 5 chickens as controls and 5 in HTL-treated group with phenotypes of NTDs. As shown in Fig. 5D, significantly higher levels of histone H3K79Hcy at E5 were detected in HTL-treated chicken with phenotypes of NTDs.

9. Lines 248-249: The authors state that H3K79Hcy-containing nucleosomes reduce NTC gene expression but upregulate the global gene expression. They also state that HTL-treatment of NE4C cells, which increases levels of H3K79Hcy, decreases occupancy of NTD-related genes by H3K79Hcy-containing nucleosomes (lines 264-269). These statements appear to be contradictory and require clarification.

Response:

We thank the reviewer for pointing out the lack of clarity in explaining the complex relationship between the level of H3K79Hcy, level of H3K79Hcy occupancy to the genes it regulates, and level of gene expression.

The following are the summary of the observation from our experiments:

- 1). H3K79Hcy level increased with an increasing level of Hcy or an increasing concentration of HTL used during treatment NE4C cells (Fig. 3B and 3C);
- 2). A decreased level of H3K79Hcy enrichment to the genes it regulates was seen during HTL treatment (Fig. 7A, 7B, and 7C);
- 3). The same trending was present for the level of H3K79Hcy enrichment and the expression

level of the genes it regulates (Fig. 6D).

Based on these observations, we concluded the following:

Treatment of NE4C cells with an increasing amount of HTL leads to an elevated level of H3K79Hcy. However, enrichment of H3K79Hcy to the genes it regulates decreases through an unknown mechanism, and subsequently results in downregulation of expression of the genes involved. Currently we are conducting experiments to define the potential mechanism.

The diagram below is intended to outline the cascade of events in a clearer fashion.

10. Lines 259-274: The findings that H3K79Hcy enrichment in NTC-related genes is reduced in response to the treatment with HTL (Fig. 7A-C) appears to be inconsistent with the finding that the treatments with HTL increase levels of N-Hcy-histone (Fig. 3C), including H3K79Hcy (Fig. S2B). The explanation of this dissonance provided in lines 405-407 of the Discussion that “higher level of H3K79Hcy has a negative impact in its binding to aforementioned genes” is not convincing.

Response:

Please see Response for Comment 9 above.

11. Lines 287-280, Fig. 7E, Table S1: The listed Hcy values are several orders of magnitude higher than published tissue Hcy values, which are about 50-100 pmol/mg in normal tissues and about 5000 pmol/mg in severe HHcy. Please check your calculations.

Response:

We thank the reviewer for the comment and wish to apologize for the error in presenting the data. The unit of Hcy level in human brain tissue should read nmol/mg, with the highest in Table S1 being 0.064 nmol/mg tissue, equivalent to 64 pmol/mg tissue.

We have made the correction in Table S1 of the revised manuscript.

12. Line 335-336: the list of N-Hay-proteins has grown substantially since 1999. See Jakubowski H. Homocysteine in Protein Structure Function and Human Disease, Springer, 2013 and more recent articles.

Response:

Thanks for the reviewer's suggestion. We have added the paper from Jakubowki et al., "Homocysteine in Protein Structure Function and Human Disease", along with some additional recent publications to the reference of the revised manuscript.

13. Line 341: The statement "...this is the first time that homocysteinylation specific to histone has been reported" is not entirely true. In fact, histone N-homocysteinylation has been examined at least in two labs: e.g., Gurda D et al., Histones are targeted for N-homocysteinylation in human endothelial cells. Acta biochimica Polonica 61 Suppl 1: 127, 2014; Xu L et al., Crosstalk of homocysteinylation, methylation and acetylation on histone H3. Analyst 140: 3057-3063, 2015.

Response:

We thank the reviewer for pointing out our error in choosing the appropriate language and are deeply impressed with the reviewer's insightful knowledge. In the revised manuscript, the sentence now reads: "Although the presence of histone N-homocysteinylation has been previously reported in samples from cultured cells, to our knowledge, this is the first time that homocysteinylation specific to histone in human tissue samples has been demonstrated."

14. Methods: Description of many procedures is missing, e.g. histone extraction from tissues/cells. Other procedures are missing important details. For example, antibodies that have been generated for this study require a more detailed description/characterization. What is the nature of Mars-KO in HEK293T cells? Is it conditional? How do you grow/treat these cells for experiments? Without these details, the work would be difficult to reproduce.

Response:

We thank the reviewer for pointing out the lack of some critical information in the previously submitted manuscript. We have followed the reviewer's advice to incorporate the missing information in the revised manuscript. Details for methods on histone extraction, development of Hcy antibody, generation of stable MetRS Knockdown cells, and construction of MetRS overexpression plasmid, are now included in the revised manuscript.

The MetRS-KO HEK293T cell is a HEK293T cell line with stable MetRS knockdown. The cell line can stably produce shRNA to inhibit MetRS expression with an inhibition efficiency

of 90%.

The MetRS knockdown HEK293T cells were cultured in DMEM, supplemented with 10% (vol/vol) FBS and 1mg/ml puromycin.

In this paper we compared histone Hcy level in MetRS knockdown HEK293T cells with that in normal cultured HEK293T cells. The results show that the decreased expression of MetRS resulted in a decreased histone Hcy level. There was no treatment of cells in this line of experimentation.

We are deeply sorry for the annotation error which may have caused some confusion in the previously submitted manuscript. We have modified the annotation about Fig. S1B.

15. If the N-Hcy-histone modification is as abundant as Zhang et al. report, it should be easily identified and quantified by another method, e.g., a direct chemical assay for Hcy in brain histones, which would greatly strengthen the authors' conclusions.

Response:

The reviewer's suggestion is well received.

In the current study, a combination of mass spectrometry and western blotting analysis has been used because these are the most sensitive and reliable methods for the detection and identification of N-Hcy-histone modification. We continue to work on generating N-Hcy-histone specific antibodies with improved sensitivity, with the goal in mind to develop chemiluminescence based N-Hcy detection kit for brain histone. In addition, we have started collaboration with analytical chemists for the development of a direct chemical assay for the detection and quantification of N-Hcy in brain histone, as suggested by the reviewer. Such tools will bring our discovery of N-Hcy in brain histone to potential clinical applications.

1. Olsen, J.V., Ong, S.E. & Mann, M. Trypsin cleaves exclusively C-terminal to arginine and lysine residues. *Mol Cell Proteomics* **3**, 608-14 (2004).
2. Rodriguez, J., Gupta, N., Smith, R.D. & Pevzner, P.A. Does trypsin cut before proline? *J Proteome Res* **7**, 300-5 (2008).
3. Ong, S.E., Mittler, G. & Mann, M. Identifying and quantifying in vivo methylation sites by heavy methyl SILAC. *Nat Methods* **1**, 119-26 (2004).
4. Burke, M.C. *et al.* Unexpected trypsin cleavage at ubiquitinated lysines. *Anal Chem* **87**, 8144-8 (2015).
5. Burke, M.C., Oei, M.S., Edwards, N.J., Ostrand-Rosenberg, S. & Fenselau, C. Ubiquitinated proteins in exosomes secreted by myeloid-derived suppressor cells. *J Proteome Res* **13**, 5965-72 (2014).
6. Maile, T.M. *et al.* Mass spectrometric quantification of histone post-translational modifications by a hybrid chemical labeling method. *Mol Cell Proteomics* **14**, 1148-58 (2015).
7. Coetzee, N. *et al.* Quantitative chromatin proteomics reveals a dynamic histone post-translational modification landscape that defines asexual and sexual *Plasmodium falciparum* parasites. *Sci Rep* **7**, 607 (2017).

Reviewer #2 (Remarks to the Author):

This is a very interesting paper that studies homocysteinylation on chromatin. Elevated homocysteine is a hallmark of folate, and B12 deficiencies and thus Neural Tube Defects. The authors identify elevated H3K79Hcy and other homocysteinylation in this setting and begin to study their function. This is an important study because it defines a novel mark on histones and begins to characterize the function in a disease setting which includes the development of an antibody to probe the mark.

Response:

First, we wish to express our sincere thanks for the reviewer's positive and encouraging comments. The reviewer's insightful questions and constructive suggestions have been thoroughly discussed among all authors and we have made every effort to address the reviewer's questions in the following.

Some concerns should be addressed.

1. In Figure 1, they used 10 human embryonic brain samples and ID'd 39 unique KHcy sites across the 4 histone variants. Were the majority of these overlapping between the samples? Essentially, how conserved are these 39 marks between different embryos? Also, any ideas about why there would be such tissue specificity (i.e. highly abundant in lung tissue but virtually absent in muscle)?

Response:

We thank the reviewer for raising such an insightful question. Accordingly, we performed analysis on the frequency of each of the 39 KHcy sites across the 10 human embryonic brain samples (shown below). Indeed, a number of KHcy sites were conserved among different samples, including H3K79Hcy (detected in all 10 samples), H2aK74Hcy (in 9 samples), H2aK95Hcy (in 9 samples), H2bK46Hcy (in 9 samples). However, some KHcy sites, such as H2aK15 and H2bK116, were present only in one sample.

From our data obtained from NE4C and HEK 293 cells, it appears that the level of histone homocysteinylation is regulated by Hcy level in the cells and we speculate that the same holds true in the tissues.

Previously it has been shown that different concentrations of free Hcy are observed in rat tissues frozen after death, including liver (1.31 nmol/g), kidneys (1.54 nmol/g), vcerebrum (0.57 nmol/g), cerebellum (4.62 nmol/g), heart (0.90 nmol/g), lung (1.03 nmol/g), and spleen (1.18 nmol/g) ¹. The distribution of Hcy concentration in rats is almost consistent with the level of Histone Hcy expression in human fetal tissues (Fig. 1E), suggesting that the level of histone homocysteinylation is regulated by the level of Hcy in the tissue.

Intracellular Hcy level is believed to be affected by both genetic and nutritional factors.

Mutations within the genes coding for enzymes participating Hcy metabolism are key attributes to the accumulation of Hcy in the tissues, these enzymes including methylenetetrahydrofolate reductase (MTFHR), methionine synthase (MS), and cystathionine-β-synthase (CBS)^{2,3}. Previously it has been reported that among human tissues, CBS mRNA levels are higher in the muscle. It is known that CBS is the key enzyme for Hcy metabolism. Patients with CBS deficiency have plasma Hcy levels of 200–300μM, compared to ~10μM in healthy individuals⁴. We speculate that level of high CBS in muscle tissue causes low Hcy levels leading to a decreased level of histone homocysteinylation in muscle. We have conducted a thorough literature search but were unable to identify any relevant publication pertaining the abundance of Hcy in the lung. With the advance in research in this field, we believe there will be emerging information to address the question the reviewer raised, as well as provide evidence of translational significance.

The histone homocysteinylation distribution of 10 human fetal brain samples

Protein Name	Modification Site	1	2	3	4	5	6	7	8	9	10	
H2A	K9	○	●	●	○	●	○	○	○	○	○	
	K15	○	●	○	○	○	○	○	○	○	○	
	K74	●	●	●	●	●	●	●	●	○	●	
	K75	○	●	●	○	●	○	●	●	○	●	
	K95	●	●	●	●	●	●	●	●	○	●	
	K99	●	●	○	●	●	●	●	●	○	●	
	K118	●	●	○	●	●	●	●	●	○	○	
	K119	●	●	●	○	●	●	●	●	○	○	
	H3	K9	●	●	●	○	●	●	●	○	○	○
		K18	●	●	○	○	●	○	○	○	○	○
K23		●	●	○	○	●	○	○	○	○	○	
K27		●	●	○	○	●	○	○	○	○	○	
K36		○	●	●	●	●	○	○	○	○	○	
K37		○	●	●	●	●	○	○	○	○	○	
K79		●	●	●	○	●	○	○	○	○	○	
K115		○	●	○	○	○	○	○	○	○	○	
H4		K8	○	●	○	○	○	○	○	○	○	○
		K16	●	●	○	○	○	○	○	○	○	○
	K31	○	●	○	○	○	○	○	○	○	○	
	K44	○	●	○	○	○	○	○	○	○	○	
	K59	●	●	○	○	○	○	○	○	○	○	
	K77	○	●	●	●	●	○	○	○	○	○	
	H2B	K5	○	●	●	○	○	○	○	○	○	○
		K11	●	●	●	○	●	●	○	○	○	○
K12		●	●	●	○	●	○	○	○	○	○	
K15		○	○	●	○	●	○	○	○	○	○	
K16		○	○	○	○	○	○	○	○	○	○	
K20		●	○	○	○	○	○	○	○	○	○	
K23		●	○	○	○	○	○	○	○	○	○	
K24		●	○	○	○	○	○	○	○	○	○	
K34		●	○	○	○	○	○	○	○	○	○	
K43		●	●	●	●	●	●	○	○	○	○	
K46	●	●	●	●	●	●	○	○	○	○		
K57	●	●	●	●	●	○	○	○	○	○		
K108	○	○	○	○	○	○	○	○	○	○		
K116	○	○	○	○	○	○	○	○	○	○		
K120	○	○	○	○	○	○	○	○	○	○		
K125	○	○	○	○	○	○	○	○	○	○		

●: the modification in this peptide was detected;
○: the modification in this peptide not detected ;

2. Assuming adult human brain samples were used in Fig. 1D, how would the authors predict H3K79Hcy expression patterns to differ between adult vs embryonic samples? Is this mark conserved into adulthood, and does the localization change?

Response:

This is an interesting question.

As a matter of fact, human fetal brain samples were used in Fig.1C and Fig.1D in the previously submitted manuscript. We agree with the reviewer that the biological roles of histone H3K79Hcy will be further elaborated if a comparison in expression patterns between adult and embryonic samples can be achieved. Currently, it is quite a challenge for us to obtain sufficient suitable samples from adult human brain, but we are making a great deal of effort to pursue that route.

From our chicken model experiments, it appears that the level of H3K79Hcy decreases

during chicken embryonic development (see Fig. 5E in the revised manuscript). It will be interesting to understand the mechanism underline this phenomenon and its translational relevance.

3. The authors looked at differences in localization between H3K4me3 (mostly localized to gene promoters) and H3K79Hcy (mostly localized to gene bodies), but does this mark co-localize with other activational epigenetic marks that typically localize in the gene body? How does the localization of typically repressive marks (like H3K27me3) compare to the localization of H3K79Hcy in these neural development genes?

Response:

We thank the reviewer for the very constructive comments.

In this manuscript, we attempted to provide evidence establishing histone homocysteinylation especially H3K79Hcy as a new epigenetic mark that may have disease implication during human embryonic development. Results from our first step mechanism studies revealed that there was a difference in gene localization between H3K4me3 and H3K79Hcy. Further studies have been designed to evaluate the potential mechanisms of H3K79Hcy on gene regulation with regard to gene localization, regulatory activity, as well as transcriptome complex, and will be conducted in reference to other hallmark epigenetic marks including H3K4me3 and H3K27me3.

Our results from NE4C cells treated with HTL indicate that while there is a global increase in H3K79Hcy, a decrease in enrichment of H3K79Hcy to the genes it regulates was observed, along with a decrease in expression of the genes involved. Such a parallel trending between H3K79Hcy enrichment and level of gene expression it regulates may suggest the role of H3K79Hcy as a transcriptional activator for the genes it regulates. However, we are precautious in interpreting such preliminary data and believe that more experiments are needed on gene body localization of H3K79Hcy, mechanism of activation, as well as co-localization with other marks, as well as whether global hyperhomocysteinylation may induce an inhibitory mechanism, preventing the binding of H3K79Hcy to its targets.

4. In Fig 4B, the authors claim that increases in H3K79Hcy don't affect methylation or acetylation, but it looks to me like acetylation does go down. Maybe this isn't important. Also, in Fig 4G, the authors say that methylation of H3K79 did not significantly change, but the panel shows a significant difference in H3K79me1 (and they didn't look at H3K79me3 or H3K79ac).

Response:

Thanks for the reviewer's comments.

Per reviewer's suggestion, we conducted a more in depth analysis on data generated from NE4C cells treated with increasing concentration of HTL. We completely agree with

the reviewer's assessment that treatment with HTL in cultured NE4C cells leads to increases in H3K79Hcy, along with effects on H3K79 methylation and acetylation. Therefore, we have modified this sentence in the revised manuscript to "And treatment with HTL in cultured NE4C cells also have effects on histone methylation and acetylation (Fig. 4D)". The sentence related to Fig. 4G has been amended to "Compared to H3K79Hcy, level of other histone modifications, i.e. methylation on H3K79 (EIAQDFKme1TDLR) increased following 0.5mM HTL treatment (Fig. 4G). Level of dimethylation on H3K79 (EIAQDFKme2TDLR) did not change significantly following HTL treatment (Fig. 4H)."

Combining data from WesternBlot analysis using anti-lysine methylation and anti-lysine acetylation antibodies (Fig. 4D), and that from mass spectrometry (Fig. 4E-H), it is clear that during HTL treatment of NE4C cells, while a significant increase in H3K79Hcy is evident (Fig. 4D-F), a downward trend in H3K79 acetylation is also observed (Fig. 4D), along with a substantial elevation in H3K79me1 although the overall H3K79 methylation (Fig.4D) and H3K79me2 (Fig.4H) levels remain largely unchanged.

5. The authors focus the majority of the paper on H3K79Hcy, which they provide sufficient rationale for. However, previous work by the authors showed that aberrant H3K79me2 results in NTD, and the authors don't show any link between these two marks in this paper. Which do they think plays a larger role?

Response:

We truly appreciate the reviewer for raising this elegant question.

The etiology of NTDs involves multiple genetic, nutritional, and environmental factors. In our previous work, we have demonstrated a potential mechanism of folate deficiency induced NTDs through attenuation of H3K79 dimethylation and subsequent aberrant expression of downstream target genes, some of which are NTDs associated. The current study focuses on the effect of an elevated level of H3K79Hcy on target gene expression that are critical for embryonic development. These are two independent studies and based on the data sets generated so far, it is difficult to conclude which modification on the same site would have a more profound role on NTDs onset. It is well understood that both folate and Hcy participate in methionine metabolism, and therefore it is expected that folate deficiency and high Hcy may cross talk in the metabolic pathways. Further studies with elegant design and larger cohorts are needed to delineate specific attributes in NTDs with H3K79 modifications.

Although not directly related to the reviewer's comments, limited mass spectrometry data from HTL treated NE4C showed that while the level of H3K79Hcy increased (Fig.4E and F), the level of H3K79me2 decreased but there was no statistical difference between the normal control and the HTL treated NE4C (Fig.4H). Based on this, it seems that 0.5mM HTL treatment in NE4C have no significant influence on H3K79me2 level.

6. If Hcy supplementation results in an overall increase in histone homocysteinylation but this mark is significantly reduced in neural development genes, where do they think this increase is occurring? Do they think it's localizing to other genes?

Response:

We thank the reviewer for raising the question.

Indeed, both histone homocysteinylation and H3K79Hcy level increased with an increasing level of Hcy or an increasing concentration of HTL used (Fig. 3B, 3C, and 4D), while a decreased level of H3K79Hcy enrichment to the genes it regulates was seen during HTL treatment (Fig. 7A, 7B and 7C). From our limited data set, it is difficult to compare the overall increased level of histone homocysteinylation to a reduced level of H3K79Hcy enrichment in neural development genes. However, the increase in global H3K79Hcy expression and the decrease in enrichment of H3K79Hcy to the genes it regulates, suggests that an unknown mechanism might be triggered under high Hcy conditions, either through an enhanced binding of H3K79Hcy to other genes as suggested by the reviewer, or a structural alteration leading to an impairment in enrichment with antibody.

7. Somewhat minor, but if MetRS is the enzyme responsible for converting homocysteine to HTL, why would MetRS knockout blunt the effects of HTL supplementation on histone homocysteinylation? (Fig S1B)

Response:

Thanks for the reviewer's question.

We are sorry for the fact that the description for Fig.S1B is misleading. The MetRS-KO HEK293T cell, as a matter of fact, is a HEK293T cell line with stable MetRS knockdown. The cell line can stably produce shRNA to inhibit MetRS expression with an inhibition efficiency of 90%.

Fig.S1B shows histone homocysteinylation level in MetRS knockdown HEK293 cells. In this particular experiment, we did not treat cells with HTL. Under MetRS knockdown conditions, Hcy would not be fully converted to HTL. Therefore, HTL level would decrease, leading to the decrease in the level of histone homocysteinylation.

We have corrected errors in the revised manuscript.

1. Svoldal, A., Refsum, H. & Ueland, P.M. Determination of in vivo protein binding of homocysteine and its relation to free homocysteine in the liver and other tissues of the rat. *J Biol Chem* **261**, 3156-63 (1986).
2. Blom, H.J., Shaw, G.M., den Heijer, M. & Finnell, R.H. Neural tube defects and folate: case far from closed. *Nat Rev Neurosci* **7**, 724-31 (2006).
3. Mills, J.L. *et al.* Homocysteine metabolism in pregnancies complicated by neural-tube defects. *Lancet* **345**, 149-51 (1995).
4. Chen, N.C. *et al.* Regulation of homocysteine metabolism and methylation in human and mouse tissues. *FASEB J* **24**, 2804-17 (2010).

Reviewer #3 (Remarks to the Author):

This is an interesting paper that suggests a molecular mechanism for the well known observation that high levels of homocysteine (Hcy) in maternal blood is a risk factor for neural tube defects in the offspring. The manuscript goes from general analysis of homocysteinylation of histones, to showing a connection between Hcy injection and NTDs in an embryonic chick model. Finally human fetal brain samples with high levels of Hcy are shown to have increased levels of H3K79Hcy and on average have decreased expression of 3 NTD genes.

1. On the whole this study is well done and represents important information. However, it is still correlative to human NTD causation and looks at Hcy levels at a time long after neurulation has occurred. It will be interesting to use the chick model to look at the effect of homocysteinylation at the time of neurulation.

Response:

Thanks to the reviewer for the suggestion and the very positive and encouraging comments.

We agree with the reviewer's comment that it is still a correlative to human NTDs causation since this is the first report to connect histone homocysteinylation to NTDs. Built on evidence present in the manuscript, we are designing and conducting a series of experiments in both animal and cellular models to explore the underline mechanism and key attributes for NTDs onset caused by histone homocysteinylation, at the levels of genetic, environmental, as well as metabolic.

In the previously submitted manuscript, we only looked at Hcy level at E5 in a chicken model due to some technical issues with sample collection and handling. In the revised manuscript, we included some new data from experiments designed to address the reviewer's concerns. The levels of H3K79Hcy during chicken brain development from E1 (24h incubation) to E5 (120h incubation) were compared between the control group (n=3) and HTL injection group (Fig. 5E). The results showed that there was an increase in histone H3K79Hcy expression during brain development in high-HTL-treated chickens while a decreased expression of H3K79Hcy was detected in samples from the normal group. Our results suggested that abnormal H3K79Hcy expression may lead to the occurrence of NTDs in chicken.

Additionally, there are places where important detail is lacking, and others where I have questions about some interpretations of Figures.

There are several places where more rational or information needs to be give:

2. Page 6, Fig 3 – Why were additional experiments only done with H3, when all 4 histones had Hcy sites mapped?

Response:

Thanks for the question raised by the reviewer.

Although Hcy sites were detected in all 4 histones, however, it has been shown previously that epigenetic marks on different histones may have different biological effects.

Results from works by Rada-Iglesias et al. have demonstrated that some histone modification sites such as H3K4me1 or H3K27ac play a key role during differentiation of human embryonic stem cells (hESCs) to neuroepithelium¹. Others have shown that H3K9me2 plays an important role on neuronal differentiation, while some of the modifications on histone H3 such as H3K9 methylation and H3K4 methylation are involved in the nervous system disease²⁻⁴. The above-mentioned three lines of evidence support the hypothesis that modification of histone H3 is more related to the development of the nervous system, relevant to NTDs. Therefore, we have been focusing our research on modification of histone H3.

3. Why was H3K79Hcy chosen for further study, over the other 5 sites that also were homocysteinyllated in untreated NE4C cells and fetal brain samples, although in different histones. (figure 4)

Response:

Thanks to the reviewer for raising such an excellent question.

Previous research has shown that H3K79 methylation play important roles in embryonic development⁵. Experimental data from our laboratory have demonstrated that abnormal H3K79 dimethylation results in altered expression of a number of NTC genes and may be involved in NTDs.

We focused our research on H3K79Hcy based on two considerations:

- a. Build on our past experience with H3K79, this particular important site during neural development in embryos;
- b. Further explore the interplay between H3K79 methylation and H3K79 homocysteination, two modifications at the same site in the next steps.

4 There are over 300 genes connected with NTDs in the mouse. Why were Smarca4, Cecr2, and Dnmt3b the genes chosen for further analysis? These epigenetic regulator genes were not in the ChIP-Seq list in Table S4 (without a title is it impossible to fully interpret this Table), while epigenetic regulator NTD genes Hdac4, Sirt1 and Smarcc1 are in this list and these were not followed up.

Response:

First, we wish to apologize for not clearly labeling the sub-tables in Table S4, which may have led to some confusion. In the revised manuscript, we have added the title 'Table S4, ChIP-seq peak genes identified from H3K4me3, histone-KHcy and H3K79Hcy' to Table S4 so that genes associated with each of the above three marks can be easily distinguished.

Indeed, there are over 300 genes connected with NTDs in the mouse and many of them are regulated by H3K79Hcy. Given the number of genes regulated by H3K79Hcy, as well as the fact that the expression of epigenetic regulator genes is more likely influenced by environmental changes, we initiated our study on biological function of H3K79Hcy by exploring its effects on epigenetic regulator gene expression and subsequent control of the expression of downstream genes. Among 14 epigenetic regulator genes connected with NTDs in the mouse, only *Cecr2*, *Smarca4* and *Dnmt3b* are founded in H3K79Hcy peak genes. Therefore, we focused our first set of experiments on *Smarca4*, *Cecr2*, and *Dnmt3b*.

5. Methods: Please give more details on the human embryonic brain samples in Table S1. The references given are not informative enough. How were the controls matched? Was the level of folic acid supplementation in the Moms the same in the controls vs the cases of NTDs? Should an anencephaly be included, since 10/11 cases are spinal defects?
- Also, were the 11 NTD brain samples with high Hcy levels selected from a larger group of NTD samples based on their Hcy levels? If so, how many samples were analysed to give this selected group?

Response:

Thanks for the reviewer's question.

Eleven samples with the highest Hcy level from 173 NTDs, and 10 samples from 178 controls were selected for the study. The controls were matched with gender (Female: 6 cases; male: 4-5 cases) and age (<20w: 2 cases; 20-30w: 7 cases; >30w: 1~2 cases).

It is very regrettable that we do not have the folate level data of maternal serum of these samples. However, it is worthwhile pointing out that based on the information collected on questionnaire survey for the study, none of the pregnant women from whom the samples for this study were used, being in the NTDs or control groups, received any folic acid supplementation. All these pregnant women came from the same area - the Lüliang area of Shanxi Province in northern China. We have added these details to the methods in the revised manuscript on line 465-470

As the anencephaly is one of the 11 samples with the highest level of Hcy, and anencephaly is subtype of NTDs, therefore, in this study, this anencephaly sample was included.

6. Figure 1-3 rely on an Anti-Hcy antibody to explore homocysteinylation. The only information on the antibody is that it came from Fudan University (no investigator name given). How was it confirmed that this antibody is specific? Is there a publication? The authors clearly show that their own Anti-H3K79Hcy is specific, but the anti-Hcy is a mystery.

Response:

We thank the reviewer for pointing out the lack of validating information for anti-Hcy antibody.

Initially, the information on the generation and validation of the anti-KHcy antibody was part of a separate manuscript. Now we decided to incorporate such information in the revised manuscript to strengthen our data set. In the revised manuscript, we included two additional figures, 1C and 1D, to demonstrate the specificity of the anti-KHcy antibody, as well as the description of results in lines 91-98. Detailed information for the generation of the antibody was also added to the method section.

Interpretation of results:

7. Fig 2, panel 2, showing HTL treatment of H3 is time dependent. I see no obvious difference between 2 and 6 hours. There is certainly an obvious difference between 2/6 and 14 hours. Make your description more specific – could 14 hrs be starting to show artifacts? Is there any quantitation for these Westerns?

Response:

Thanks to the reviewer for the excellent question and certainly the reviewer's rigorous scientific attitude is very worthwhile for us to learn.

The reviewer is quite right that there was no obvious difference in signal intensity between incubation for 2 and 6 hours in a dot blot using an Hcy antibody. Therefore, we have modified this sentence in the revised manuscript. Our data show that the Hcy intensity signal increased after HTL treatment with 12 hours (data not shown).

8. Line 181-182, Fig 4D – “Furthermore, treatment with HTL in cultured NE4C cells had no obvious effects on histone methylation and acetylation (Fig. 4D).” Looking at this Figure I would say that methylation goes up a little at 0.5mM and then back down again at 1 mM. Also, acetylation is considerably decreased at 1 mM. The above sentence needs to be modified. What would the significance of these differences be?

Response:

We thank the reviewer for careful reading of the manuscript.

Per reviewer's suggestion, we took a more in depth analysis on data generated from NE4C cells treated with increasing concentrations of HTL. We completely agree with the reviewer's assessment that treatment with HTL in cultured NE4C cells leads to increases in H3K79Hcy, along with effects on H3K79 methylation and acetylation. Therefore, we have modified this sentence in the revised manuscript to "And treatment with HTL in cultured NE4C cells also have effects on histone methylation and acetylation (Fig. 4D)". The sentence related to Fig. 4G has been amended to "Compared to H3K79Hcy, level of other histone modifications, i.e. methylation on H3K79 (EIAQDFKme1TDLR) increased following 0.5mM HTL treatment (Fig. 4G). Level of dimethylation on H3K79 (EIAQDFKme2TDLR) did not change significantly following HTL treatment (Fig. 4H)."

The difference in histone methylation and acetylation when cells were treated with 0.5mM and 1mM HTL indicates that there is a balance between homocysteinylation, methylation and acetylation. Abnormal expression of any one of these epigenetic marks may result in development defects including NTDs. An optional balance between modification levels of homocysteinylation, methylation and acetylation may be the key to maintain a health, normal development path, an algorithm we are envision for our future studies.

9. Line 186-188, Figure 4 G & H. "Compared to H3K79Hcy, level of other histone modifications, i.e. methylation on H3K79 did not change significantly following HTL treatment (Fig. 4G and 4H)." But Figure 4G shows a significant difference between control and HTL treatment!

Response:

Thanks for the suggestion. The sentence related to Fig 4G was modified to "Compared to H3K79Hcy, level of other histone modifications, i.e. methylation on H3K79 (EIAQDFKme1TDLR) increased following 0.5mM HTL treatment (Fig. 4G). Level of dimethylation on H3K79 (EIAQDFKme2TDLR) did not change significantly following HTL treatment (Fig. 4H)."

10. In Figure 7, I find panels A and B difficult to reconcile with panels C and D. In A (ChIP-Seq) only Cccr2 shows a large difference, but in C (ChIP-PCR) Dnmt3b shows the biggest difference. Panel B doesn't match panel d. I understand these panels compare different techniques, but some explanation should be given. Also, why is Smarca4 the only gene analysed in more depth (panel B)?

Response:

We completely agree with the reviewer's comment that an explanation is warranted for the unmatched results seen between ChIP-seq (Fig. 7A top) and ChIP-PCR (Fig. 7C), as well as between RNA-seq (Fig. 7A bottom) and RT-qPCR (Fig. 7D).

ChIP-seq and RNA-seq are genome-wide analysis capturing each fragment derived from

a specific gene or its transcript so that a relative abundance can be calculated. While CHIP-PCR and RT-PCR can generate quantitative data under optimal experimental conditions, the selection of the regions for amplification and the design of the primer sets may play a critical role in causing variation in data. In this particular study, we utilized CHIP-PCR and RT-PCR only to confirm the results obtained from two genome wide analysis tools, Chip-Seq and RNA-seq.

In this manuscript, we attempted to provide evidence establishing histone homocysteinylation especially H3K79Hcy as a new epigenetic mark that may have disease implication during human embryonic development. First step mechanism studies revealed that difference in gene localization between H3K4me3 and H3K79Hcy. Smarca4 was used as an example to illustrate the binding of H3K79Hcy in genomic location.

Other comments:

11. Please put Titles and brief explanations on all Supplementary Tables! None currently have titles.

Response:

Thanks for your suggestion. We have put Titles and brief explanations on all Supplementary Tables in the revised manuscript.

12. Introduction: What is the connection between Hcy levels and folate supplementation? If folate is supplemented, does the connection between Hcy and NTDs disappear?

Response:

We thank the reviewer for raising this simple but difficult to answer question.

The level of Hcy is influenced by many factors, and among them, folic acid level is one of the most important factors. Previous studies have shown that supplementation of folic acid can reduce the level of Hcy in serum⁶⁻⁸, and folic acid supplementation significantly reduces the incidence of NTDs^{9,10}. The reviewer's question regarding the connection between Hcy levels and folate supplementation can be best answered with the results from a study from Xiaoying Zheng's group in Shanxi, China. It shows that a community intervention was carried out in which 16,648 women of child-bearing age were supplemented with folic acid. The serum folic acid level of these pregnant women increased significantly while the level of Hcy decreased, along with a significant decrease in the incidence of NTDs¹¹.

Data from an earlier research also showed that Hcy could induce teratogenesis in chick neurulation and organogenetic period. If folate is supplemented, the teratogenicity of Hcy was significantly antagonized, the occurrence rate of NTDs was down from 43.5% to 0¹².

13. There was no mention of homocysteine in mouse models of NTDs. Bennett et al (2006) found that an increase in maternal homocysteine did not result in neural tube defects. Greene et al (2003) cultured mouse embryos in 0.5mM and higher homocysteine and saw signs of toxicity but no NTDs, concluding that high levels of homocysteine may represent a metabolic disturbance rather than something causal. Since the authors are using mouse models to identify NTD genes, there should be some comment on whether the mouse can be used as a model to study the role of homocysteine levels in NTDs. Could any of their findings be tested in mice? Or is this more evidence for the utility of the chick model?

Response:

Thanks to the reviewer for another great question.

Published reports have shown that mice treated with Hcy or HTL could not develop NTDs.

- Van Aerts et al. treated embryos on Day 10 after conception with high homocysteine concentration (4mU), but did not observe any abnormalities¹³.
- Hansen et al. also cultured mice embryo (GD 8) with Hcy-containing medium (2-200 µg/ml) or HTL-containing medium (100 µg/ml) but did not see any deformity¹⁴. Hansen et al. also injected 0.2-10µg HTL directly into the amniotic membrane of mouse embryos by microinjection, and it did not produce any deformities either.
- Greene et al. cultured embryo from CD-1 DX mice in high homocysteine medium for full embryo culture and there was no neural tube malformation¹⁵.
- Watanabe et al. established a mouse model by inactivating CBS gene to produce high homocysteine serum mutant mice. It was then used to mate with heterozygous mice to produce homocysteine concentration two times higher than normal mice. These mice had no neural tube defects¹⁶.
- Rosenquist used 7 rat species (SWV, LMBc, Dai, NHRI, CBA, Splotch, ICR) to give high concentration homocysteine once in GD 8, or to give high concentration homocysteine daily from GD 6.5 to GD 10.5, and could not induce neural tube defects either¹⁷.

Therefore, it seems that the accumulating evidence supports the notion that the treatments or the models tested so far illustrate that HTL or Hcy treatment does not cause NTDs in mouse.

However, reports in the literature show that treatment of either Hcy or HTL can quickly obtain NTDs phenotypes in chicken embryo model^{12,18,19}.

The purpose of our animal model is to obtain NTDs to verify if the H3K79Hcy is involved in causing NTDs in animal models. So we chose HTL injection chicken model as our animal model.

It has been hypothesized that the difference in roles of homocysteine causing NTDs between chicken and mice resides in differences in NMDA receptor inhibition²⁰.

14. Figure 5, chick experiment. Is 0.5 mM HTL close to the levels that human embryos might experience, or is it much larger? Did you try lower doses and still see NTDs?

Response:

We thank the reviewer for bringing up this very important question.

Our starting point for the chick experiment was to take a reference on work done by MARTA EPELDEGUI¹⁸, who used 20 μ M HTL 8 μ l in the chick embryo. And we used 0.5mM HTL with 0.5 μ l in our experiment. In our initial experimentation, we did use two different concentrations, 0.1mM and 0.5mM. At 0.1mM, chicken embryo survival rate was 100% with a malformation rate of 12.5%. Therefore, we used 0.5mM in subsequent experiments.

It is difficult to determine whether is 0.5 mM HTL close to the levels that human embryos might experience. Because there's no related data in human brain.

15. Line 74 – clarify which 10 brain samples were used (I presume the 10 normal samples in Table S1).

Response:

Thanks for the reviewer's question. Indeed, ten fetal brain samples are the 10 normal samples in Table S1. We have added the information in the revised manuscript.

16. Line 125-127 – “It is worth mentioning that the number of homocysteinylation sites seemed to correlate with the intensity of KHcy signal on western blotting (Figure 2B top panel and 2E).” H4 does not fit the pattern.

Response:

Thanks for the reviewer's carefully analysis of the data and question. We have modified this sentence to “It is worth mentioning that the number of homocysteinylation sites seemed to correlate with the intensity of KHcy signal except H4 on western blotting (Figure 2B top panel and 2E).”

17. Figure S2 – There is no description of panel C

Response:

Thanks for catching the error. We have ever since added the description to panel C.

18. Line 197 – Why is “similar NTC gene distribution on chromosomes as human embryos” an advantage for the chick system?

Response:

Thanks to the reviewer for raising the question.

The expression pattern of NTC genes in chicken is quite similar to those in human embryo, indicating the conservatism of these genes. Such a conservation may derived from the similar chromosomal localization of these genes, representing a relevant translational value of the chicken animal models.

19. Line 207: “When the same amount of HTL was injected into placenta, similar results were obtained with regarding rates of embryo survival rate, overall malformations, and occurrence of NTDs (data not shown).” Please add the numbers or remove. Please clarify – what do you mean by injecting into the chicken “placenta”.

Response:

We wish to apologize for the incorrect word used in the manuscript submitted previously. It should read blastoderm injection, or blastoderm cavity injection to be more accurate. Because this experiment is a preliminary experiment, the results have shown that the teratogenic effect is not as good as the injection of neural groove, and the number of samples is relatively small. Therefore we have removed this sentence from the revised manuscript.

20. Is Table S4 a list of ChIP-Seq-detected genes from using anti-H3K4me3 or anti-H3K79KHcy or both?

Response:

First, we wish to apologize for not clearly labeling the sub-tables in Table S4, which may have led to some confusion. In the revised manuscript, we have added the title ‘Table S4, ChIP-seq peak genes identified from H3K4me3, KHcy and H3K79Hcy’ to Table S4 so that genes associated with each of the above three marks can be easily distinguished.

21. Lines 356-359 – This is a very long and convoluted sentence that is very difficult to understand.

Response:

Thanks for your suggestion. We have modified this sentence to “Although high concentrations of HTL were used during *in vitro* and *in vivo* treatment, yet, the most KHcy sites were detected from fetal brain samples. These indicated that in the brain, histone

homocysteinylation involving cellular metabolism is far more efficient than direct chemical reactions carried out by HTL or Hcy. ”

22. Lines 387-391 – this passage in the Discussion gives new results that were not discussed in the results section, and refers to Fig. 4S. Furthermore, the list of genes given here are the ones they tested, but Fig 4S shows that not all of them showed significant decrease in expression. This section should be in the results and identify only the genes that showed an increase. Then it can be discussed in the discussion.

Response:

We thank the reviewer for pointing out the error in organizing the data. We will move this passage to the RESULTS section.

23. Line 395 – Do the authors mean “binding of H3K79-Hcy ..?”

Response:

Thanks for your question. Yes, it means “binding of H3K79Hcy”. We have modified this sentence in the revised manuscript.

24. Line 707 – name the unrelated control antibodies used

Response:

The positive control antibody we used in this experiment was a tublin antibody, diluted 1:1000. The purpose of including antibody was to show that the dot blot system worked well under experimental conditions.

25. Table 7 headings are run together and not readable. Also, the Table would be much more useful if gene name symbols were also included along with gene ID numbers. The lines are fragmented into 3 non-consecutive pages, making it very difficult to look at anything specific.

Response:

We are so sorry for the issue and believe that it was caused, most likely, by the file incompatibility between different computers because the length of the EXCELL table. We made attempts to convert it into a compatible PDF format but it was not successful because the EXCELL table is too wide. We are seeking technical advices and will also communicate with the publisher to get this resolved.

The reviewer’s suggestion on including gene name symbol in the EXCELL table is well received and has been followed upon.

26. The difference in Hcy levels in human embryo brains is very impressive in Figure 7E first panel. This leads to a wide variation in Figure 7E 3rd panel where H3K79Hcy modification is measured, with many individuals within the normal range. This could be due to many factors, considering that the Hcy levels are being measured long after neurulation has occurred, different genetic backgrounds may show different resistance to high Hcy levels, and there may be many different causes for the 11 NTDs. I wonder if the subset of individuals above the normal range of H3K79Hcy were analysed separately for panel F, if you would find a bigger difference in expression of the 3 NTD genes.

Response:

Thank you for your suggestion.

Per your suggestion, we further divided the 11 NTDs into two groups: H3K79Hcy expression level above the average of the normal group (NTDs-High H3K79Hcy, n=5) and that below the average of the normal group (NTDs-low H3K79Hcy, n=6), and compared the expression level of *Smarca4*, *Cecr2* and *Dmnt3b* between these two groups. The results show that the expression of *Smarca4* and other 2 genes is lower in NTDs-high H3K79Hcy group than that in NTDs-low H3K79Hcy group, indicating that the higher the level of H3K79Hcy expression, the lower expression of *Smarca4*, *Cecr2* and *Dmnt3b*. This confirms the reviewer's hypothesis.

However, due to the fact that the sample size for the NTDs group is relatively small in this study, we could not achieve desired statistical significance in several comparisons, therefore, the original graph in previously submitted manuscript will be used.

Reference

1. Rada-Iglesias, A. *et al.* A unique chromatin signature uncovers early developmental enhancers in humans. *Nature* **470**, 279-83 (2011).
2. Ryu, H. *et al.* ESET/SETDB1 gene expression and histone H3 (K9) trimethylation in Huntington's disease. *Proc Natl Acad Sci U S A* **103**, 19176-81 (2006).
3. Shi, Y. Histone lysine demethylases: emerging roles in development, physiology and disease. *Nature Reviews Genetics* **8**, 829-33 (2007).
4. Akbarian, S. & Huang, H.S. Epigenetic regulation in human brain-focus on histone lysine methylation. *Biol Psychiatry* **65**, 198-203 (2009).

5. Jones, B. *et al.* The histone H3K79 methyltransferase Dot1L is essential for mammalian development and heterochromatin structure. *Plos Genetics* **4**, e1000190 (2008).
6. Dehkordi, E.H., Sedehi, M., Shahraki, Z.G. & Najafi, R. Effect of folic acid on homocysteine and insulin resistance of overweight and obese children and adolescents. *Adv Biomed Res* **5**, 88 (2016).
7. Manizheh, S.M. *et al.* Comparison study on the effect of prenatal administration of high dose and low dose folic acid. *Saudi Med J* **30**, 88-97 (2009).
8. Selhub, J. & Rosenberg, I.H. Excessive folic acid intake and relation to adverse health outcome. *Biochimie* **126**, 71-8 (2016).
9. Obican, S.G., Finnell, R.H., Mills, J.L., Shaw, G.M. & Scialli, A.R. Folic acid in early pregnancy: a public health success story. *FASEB J* **24**, 4167-74 (2010).
10. Czeizel, A.E. Periconceptional folic acid and multivitamin supplementation for the prevention of neural tube defects and other congenital abnormalities. *Birth Defects Res A Clin Mol Teratol* **85**, 260-8 (2009).
11. Wang, H. *et al.* Effectiveness of Folic Acid Fortified Flour for Prevention of Neural Tube Defects in a High Risk Region. *Nutrients* **8**, 152 (2016).
12. Li, Y., Li, Z., Chen, X. & Qi, P. [Homocysteine-induced neural tube defects in chick embryos and protection of folic acid]. *Wei Sheng Yan Jiu* **27**, 372-6 (1998).
13. Vanaerts, L.A. *et al.* Prevention of neural tube defects by and toxicity of L-homocysteine in cultured postimplantation rat embryos. *Teratology* **50**, 348-60 (1994).
14. Hansen, D.K., Grafton, T.F., Melnyk, S. & James, S.J. Lack of embryotoxicity of homocysteine thiolactone in mouse embryos in vitro. *Reprod Toxicol* **15**, 239-44 (2001).
15. Greene, N.D., Dunlevy, L.E. & Copp, A.J. Homocysteine is embryotoxic but does not cause neural tube defects in mouse embryos. *Anat Embryol (Berl)* **206**, 185-91 (2003).
16. Watanabe, M. *et al.* Mice deficient in cystathionine beta-synthase: animal models for mild and severe homocyst(e)inemia. *Proc Natl Acad Sci U S A* **92**, 1585-9 (1995).
17. Bennett, G.D. *et al.* Failure of homocysteine to induce neural tube defects in a mouse model. *Birth Defects Res B Dev Reprod Toxicol* **77**, 89-94 (2006).
18. Kobus-BianchiniEpeldegui, M., Pena-Melian, A., Varela-Moreiras, G. & Perez-Miguelsanz, J. Homocysteine modifies development of neurulation and dorsal root ganglia in chick embryos. *Teratology* **65**, 171-9 (2002).
19. Kobus-Bianchini, K., Bourckhardt, G.F., Ammar, D., Nazari, E.M. & Muller, Y.M. Homocysteine-induced changes in cell proliferation and differentiation in the chick embryo spinal cord: implications for mechanisms of neural tube defects (NTD). *Reprod Toxicol* (2017).
20. Bennett, G.D., Moser, K., Chaudoin, T. & Rosenquist, T.H. The expression of the NR1-subunit of the NMDA receptor during mouse and early chicken development. *Reprod Toxicol* **22**, 536-41 (2006).

Reviewers' Comments:

Reviewer #1:

Remarks to the Author:

The authors have satisfactorily responded to most, but not all, of my comments. A few loose ends, indicated below, still remain to be tied up.

Major points:

Introduction

1. Lines 49-51: In its present form, the sentence "Results from several other studies..." is misleading because it cites only one of the two prior studies and suggests that other studies have been carried out only in HEK293T cells, while in fact human umbilical vein endothelial cells (HUVECs) have also been studied (ref: Gurda, D., Marczak, L. & Jakubowski H. Histones are targeted for N-homocysteinylation in human endothelial cells. *Acta Biochim Pol* 61 Suppl 1, 127 (2014) [http://www.actabp.pl/pdf/Supl1_14/Session_6.pdf]. This reference should be cited and the sentence should be corrected to read "Results from two other studies have demonstrated the presence of histone homocysteinylation in HEK283T9 and human endothelial cellsRef, however, there have been no published report on histone homocysteinylation in human fetal tissue."

Results

2. Line 329, Table S1: The Hcy values expressed in units of nmol/mg are awkward. Please change the units to pmol/mg and reduce the number of decimal points in each value shown. The reported Hcy values, corresponding to 4.1 μ M and 0.3 μ M in NTDs and controls, respectively, and are now suspiciously low. Perhaps this is due to an unreferenced Hcy assay used by the authors, details of which are missing in the Methods on lines 659-663.
3. Fig 7E, Lines 936, 937: The Hcy values shown in the legend to Fig. 7E are missing units. The units on the y axis in Fig 7E are still incorrect.
4. Line 728: Ref. 18 is incorrect to support the statement on lines 112, 113. The correct citation is Jakubowski, H. Metabolism of Homocysteine Thiolactone in Human Cell Cultures. *J Biol Chem* 272, 1935-1942 (1997).
5. Fig. 1(A), lines 806, 807, and Table 1, lines 84, 85: Not clear which histone homocysteinylation sites were identified in the human NTD brains and which in controls. Please clarify.
6. Fig. 2(E), Line 145, line 855, and Fig. 3(D), line 868: Please indicate which brain samples are meant here, NTD or control?

Methods

Although the Methods section has been somewhat expanded in response to my comments, important details of Hcy assays and mass spectrometry analyses are still missing.

7. Line 201: the authors state that "The level of H3K79Hcy was also quantified using a mass spectrometry label-free method". However, they do not provide any information regarding how exactly these analyses were carried out, which one would expect to be included in the Methods.
8. Fig. 4E-H: Please state in the legend/Methods how the Hcy-peptide quantification was done.
9. Line 602: Identification of PTMs generally requires enrichment for the PTP-containing peptides. As N-homocysteinylation is not a stoichiometric modification, one can expect that the enrichment for N-Hcy-peptides would be required, yet the authors do not provide any information to that effect. Please state in the Methods section whether the tryptic digests were enriched in Hcy-peptides, and if so, how?
10. Lines 659-663: The description of a mass spectrometry assay for Hcy is inadequate and not informative. The second sentence of the two-sentence description doesn't make sense. A brief and informative description supported by a reference should be provided.

Minor points, typos:

Line 829: Should be 'Direct', not 'Direction'

Line 834: Should be 'tubulin', not 'tublin'

Fig. 4B, C, D: Should be 'Ponceau stain', not Poneau stain'

Reviewer #2:

Remarks to the Author:

The authors have provided satisfactory responses to my questions. This will be an important paper.

Reviewer #3:

Remarks to the Author:

The authors have produced a much improved version of their manuscript. They have added critical information in their Methods section, particularly with respect to the characterization of their 2 antibodies. They have given explanations for the reviewers questions, and in several cases I think those explanations should be included in the manuscript – see below. This will be an important publication in the field.

From Reviewers 3

#1 - I understand that it is impossible to test Hcy level in human neurulating samples. However, the fetuses used were almost all in at least the second trimester, when much brain development and differentiation has occurred, long after neurulation. Also, the NTDs in these fetuses was spina bifida, which affects very early spinal cord, rather than brain. The Hcy conditions and histone homocysteinylation could be different at the time of neurulation. I think that it is necessary for the authors to point out these facts as a possible caveat. They could then point to the new chick data and comment on it in that context.

#2 – Nice explanation of why modifications on H3 are the focus of this manuscript. Please include this brief explanation in the manuscript.

#4 – Again, please add the explanation to the manuscript.

#5 – Thanks for adding the human samples details requested to the manuscript. I do feel that the anencephalic fetus should be excluded. Brain development is very different in an anencephalic fetus compared to normal development and the disorganized brain tissue is also open to the amniotic fluid (and thus at some point degrading). Any differences in brain samples could be related to this and therefore skew the results. The sample is still included in the manuscript, but not mentioned (it is in a reference). In fact, on line 458-459 the text specifies that the fetuses were diagnosed with spina bifida. If the anencephalic sample is included, that must be stated in the methods and results. Yes exencephaly and spina bifida are both NTDs, but there are also major differences.

#13 – When describing the chick model, it would be good to briefly mention that increases in Hcy or HTL do not lead to NTDs in mice. Since so much work has been done on mice and the KO mice for the 3 genes focused on are described in mice, I think it's important to mention. It can be stated this is one of the reasons a chick model was used.

From reviewer 2, response to question #1 – the authors have produced an interesting figure which could be included as supplementary data.

In several places (line 220 and Fig 5 Legend, line 899-900), the authors discuss treatment on chicks after they are "hatched". Do they mean after the egg is laid? Hatching is long after neurulation.

Figure 1 – the added quantitation of the Western blot in 1c needs to be described in the legend.
1f – are all species samples brain tissue? Fetal brain tissue?

Line 217 “including similar NTC gene distribution on chromosomes as human embryos???” This was explained in the authors rebuttal, but it needs to be explained further in the manuscript.

REVIEWERS' COMMENTS:

Reviewer #1 (Remarks to the Author):

The authors have satisfactorily responded to most, but not all, of my comments. A few loose ends, indicated below, still remain to be tied up.

Response:

Thanks to the reviewer for the suggestion and the very positive and encouraging comments. The reviewer's questions and suggestions have been thoroughly discussed among all authors and we addressed the reviewer's questions in the following.

Major points:

Introduction

1. Lines 49-51: In its present form, the sentence "Results from several other studies..." is misleading because it cites only one of the two prior studies and suggests that other studies have been carried out only in HEK293T cells, while in fact human umbilical vein endothelial cells (HUVECs) have also been studied (ref: Gurda, D., Marczak, L. & Jakubowski H. Histones are targeted for N-homocysteinylation in human endothelial cells. Acta Biochim Pol 61 Suppl 1, 127 (2014) [http://www.actabp.pl/pdf/Supl1_14/Session_6.pdf]. This reference should be cited and the sentence should be corrected to read "Results from two other studies have demonstrated the presence of histone homocysteinylation in HEK283T9 and human endothelial cellsRef, however, there have been no published report on histone homocysteinylation in human fetal tissue."

Response:

Thanks for the reviewer's question.

We are very thankful to the reviewer for the insightful suggestion to improve the sentence in lines 50-53. The reference about the histone N-homocysteinylation study in human endothelial cells has been cited and according to the reviewer's suggestion, now the sentence reads as "Results from two other studies have demonstrated the presence of histone homocysteinylation in HEK293T (Xu et al. 2015) and human endothelial cells(Gurda, Marczak, and Jakubowski 2014), however, there have been no published report on histone homocysteinylation in human fetal tissue."

Results

2. Line 329, Table S1: The Hcy values expressed in units of nmol/mg are awkward. Please change the units to pmol/mg and reduce the number of decimal points in each value shown. The reported Hcy values, corresponding to 4.1 μ M and 0.3 μ M in NTDs and controls, respectively, and are now suspiciously low. Perhaps this is due to an unreferenced Hcy assay used by the authors,

details of which are missing in the Methods on lines 659-663.

Response:

Thanks for the suggestion. We have changed the nmol/mg to pmol/mg to reduce the number of decimal points in Supplementary Table 1. In the revised manuscript, the Hcy values in this paper are 40.6 pmol/mg brain in NTDs and 3.3 pmol/mg brain in normal controls. Since the reviewer 3 suggested the anencephalic fetus should be excluded, we re-calculated the Hcy values in revised paper accordingly and now they are 41.0 pmol/mg brain in NTDs and 3.3 pmol/mg brain in normal controls, respectively. Based on Hcy distribution studies in mammalian, hcy concentrations have been found to be 1.36-10.74 pmol/mg, 0.25-1.07 pmol/mg, 0.49-6.40 pmol/mg, 0.34-1.09 pmol/mg in the brain of rabbit, guinea pig, rat and mouse, respectively (Broch and Ueland 1984). Thus, the Hcy level of these brain is in the range of 0.25-10.74 pmol/mg. The value of 3.3 pmol/mg brain in normal controls detected in this study is within this range.

The method for analyzing Hcy level is based on protocol developed in our group which was published recently (Zhang et al. 2018). We have included a detailed description of the methods as the following: "Hcy level detection is set up by our laboratory. Brain tissue were treated with 150 mL of 50 mM DTT and waited a 20 min period in room temperature to reduce disulfide bonds, then 200 uL of internal standard (Hcy-d4) were added. Spiked brain samples were vortexed and homogenized before a 15 min sonication and a 12000g centrifuge. The supernatant were transferred to a solid phase extraction (SPE) tip in a commercial kit named EZ:faast, sample purification and Hcy derivatization was conducted according to the manufacturer's protocol. After that, the derivative Hcy was evaporated and re-dissolved using methanol-water (65:35, v/v) containing 1 mM ammonium formate before injection. An Agilent 6410B triple-quadrupole mass spectrometer with an Agilent 1200 system HPLC (Palo Alto, CA, USA) were used for LC-MS/MS analysis. Separation was performed on a Zorbax Bonus-RP column (100 mm*2.1 mm i.d., 1.8 mm particle size, Agilent Technologies, Germany) at a flow rate of 0.25 mL/ min. The mobile phase was methanol - water (65:35, v/v) containing 1 mM ammonium formate. Each sample was injected in a volume of 1 mL via an auto-sampler and separated by isocratic elution in 6.5 minutes. The column temperature was 35°C. The MS/MS experiments were performed under positive-ion (ESI+) mode with multiple-reaction monitoring (MRM). The capillary voltage was set to 4 kV and the source temperature was set to 350°C. Nitrogen served as the nebulizer gas at a flow rate of 10 L min/1 and a pressure of 45 psi. High purity nitrogen was used as the collision gas. The MRM transition for Hcy and Hcy-d4 were 350-204.1 and 354.2-208.1, respectively."

3. Fig 7E, Lines 936, 937: The Hcy values shown in the legend to Fig. 7E are missing units. The units on the y axis in Fig 7E are still incorrect.

Response:

Thanks for the reviewer's question. We have modified Fig.7E in revised manuscript. As the reviewer 3 suggested the anencephalic fetus should be excluded. The new Fig. 7E will just include 10 normal controls and 10 NTDs.

4. Line 728: Ref. 18 is incorrect to support the statement on lines 112, 113. The correct citation is Jakubowski, H. Metabolism of Homocysteine Thiolactone in Human Cell Cultures. J Biol Chem 272, 1935-1942 (1997).

Response:

Thanks for the reviewer's suggestion. We have corrected the error in the reference and included the right reference.

5. Fig. 1(A), lines 806, 807, and Table 1, lines 84, 85: Not clear which histone homocysteinylation sites were identified in the human NTD brains and which in controls. Please clarify.

Response:

Thanks for the reviewer's suggestion. Fig.1(A) and Table 1: all these histone homocysteinylation sites were identified from the 10 normal human brains. Preliminary data showed that there was no new histone homocysteinylation sites detected in NTD samples. We have modified these two sentences in the revised manuscript as below:

Fig.1(A), which was changed position to Fig1(B) in the revised manuscript : Schematic illustration of homocysteinylation sites of histone lysine residues in human normal brain samples identified using HPLC-MS/MS.

Table 1: Altogether, 39 histone KHcy sites were identified in four major histone variants from 10 normal controls.

6. Fig. 2(E), Line 145, line 855, and Fig. 3(D), line 868: Please indicate which brain samples are meant here, NTD or control?

Response:

Thanks for the reviewer's suggestion. Fig. 2(E), Line 145, line 855, and Fig. 3(D), line 868: are all normal human fetal brain. We have modified these three sentences in revised manuscript as below:

1. Fig. 2(E): Interestingly, 19 histone KHcy sites were also found in normal human fetal brain samples (depicted with red dot in Fig. 2E).
2. Line 921: Homocysteinylation sites, present both naturally in normal human brain samples (Fig. 1B) and after in vitro HTL treatment are marked with a red dot.
3. Fig. 3(D): Homocysteinylation sites detected in both normal fetal brain (Fig. 1B) and NE4C are marked with red dots under numbering of lysine residues.

Methods

Although the Methods section has been somewhat expanded in response to my comments, important details of Hcy assays and mass spectrometry analyses are still missing.

Response:

Thanks for the reviewer's suggestion. We have made relevant supplements and modifications according to your opinion. The details see as the respond to question 7, 8 and 10.

7. Line 201: the authors state that "The level of H3K79Hcy was also quantified using a mass spectrometry label-free method". However, they do not provide any information regarding how exactly these analyses were carried out, which one would expect to be included in the Methods.

Response:

Thanks for the reviewer's suggestion. We agree with the reviewer's comment that it would be beneficial for the general readers to include a more detailed description on label-free method used for the quantification of level of H3K79Hcy. The label-free method we used in our paper was the PRM method did not explain clear. We have modified the sentence on line 173 "To further corroborate these observations, label-free quantitative mass spectrometry (PRM: parallel Reaction Monitoring) was used to identify histone sites before and after pretreatment of HTL." We have also modified the sentence on line 207 as "The level of H3K79Hcy was also quantified using a mass spectrometry label-free (PRM) method and Skyline software." We also confirm the skyline detected results by area calculation of the raw data showed as supplementary figure 2G.

8. Fig. 4E-H: Please state in the legend/Methods how the Hcy-peptide quantification was done.

Response:

Thanks for the reviewer's suggestion. In the revised manuscript, Lines 626-642 now include "PRM" and "PRM data analysis" describing the methods how the Hcy-peptide quantification has been carried out.

9. Line 602: Identification of PTMs generally requires enrichment for the PTP-containing peptides. As N-homocysteinylation is not a stoichiometric modification, one can expect that the enrichment for N-Hcy-peptides would be required, yet the authors do not provide any information to that effect. Please state in the Methods section whether the tryptic digests were enriched in Hcy-peptides, and if so, how?

Response:

Thanks for the reviewer's question.

We agree with the reviewer's comment that identification of PTMs generally requires enrichment

for the PTP-containing peptides, and there have been abundance of precedents (Du et al. 2015; Zhao et al. 2010; Xie et al. 2016). The reviewer's comment that N-homocysteinylation is not a stoichiometric modification and may require enrichment the Hcy-containing peptides is well received. During the initial set up for this study, we followed previously published methods (Marczak et al. 2011; Xu et al. 2015) which demonstrated the results using a protocol without enrichment with a specific antibody. Preliminary results from mass spectrometry in our experiments showed that both the coverage rate of peptides, and the detection rate of modification are reasonably acceptable for the purpose of this study. Thus in our study, we have used the method without enrichment with specific antibodies. For our future experimentation, we plan to first evaluate the two methods head-to-head, and incorporate the enrichment step to improve the detection rate as needed.

10. Lines 659-663: The description of a mass spectrometry assay for Hcy is inadequate and not informative. The second sentence of the two-sentence description doesn't make sense. A brief and informative description supported by a reference should be provided.

Response:

Levels of Brain Hcy were determined using a LC-MS/MS method from a recent publication in our group (Zhang et al. 2018). In brief, brain tissue was first homogenized, sonicated and centrifuged, followed by the solid phase extraction (SPE) and derivatization using the EZ:faast kit. After derivatization, sample was evaporated and re-dissolved before injection. An Agilent 6410B mass spectrometer (ESI+) with an Agilent 1200 system HPLC (Palo Alto, CA, USA) and a Zorbax Bonus-RP column set at 35°C (100 mm*2.1 mm i.d., 1.8 mm particle size, Agilent Technologies, Germany) were used for LC-MS/MS analysis. Mobile phase was methanol/water (65/35, v/v) containing 1 mM ammonium formate with a flow rate of 0.25 mL/ min. Each sample was injected in a volume of 1 mL and separated by isocratic elution. The MRM transition for Hcy and Hcy-d4 were 350-204.1 and 354.2-208.1, respectively.

Minor points, typos:

Line 829: Should be 'Direct', not 'Direction'

Line 834: Should be 'tubulin', not 'tublin'

Fig. 4B, C, D: Should be 'Ponceau stain', not 'Poneau stain'

Response:

Thanks for the reviewer's catch. We have corrected these errors in the revised manuscript.

Reviewer #2 (Remarks to the Author):

The authors have provided satisfactory responses to my questions. This will be an important paper.

Response:

First, we wish to express our sincere thanks for the reviewer's positive and encouraging comments. The reviewer's questions and suggestions have been thoroughly discussed among all authors and we have made every effort to address the reviewer's questions in the following.

--

Reviewer #3 (Remarks to the Author):

The authors have produced a much improved version of their manuscript. They have added critical information in their Methods section, particularly with respect to the characterization of their 2 antibodies. They have given explanations for the reviewers questions, and in several cases I think those explanations should be included in the manuscript – see below. This will be an important publication in the field.

From Reviewers 3

#1 - I understand that it is impossible to test Hcy level in human neurulating samples. However, the fetuses used were almost all in at least the second trimester, when much brain development and differentiation has occurred, long after neurulation. Also, the NTDs in these fetuses was spina bifida, which affects very early spinal cord, rather than brain. The Hcy conditions and histone homocysteinylation could be different at the time of neurulation. I think that it is necessary for the authors to point out these facts as a possible caveat. They could then point to the new chick data and comment on it in that context.

Response:

Thanks for the reviewer's suggestion. We have point these limitations in the discussion in the revised manuscript as below:

“It is possible that the Hcy level and histone homocysteinylation level presented in this study may not accurately reflect that of neural tube closure at the time of neurulation because the fetuses used in this study were in at least the second trimester, long after neurulation, and NTD samples in this study were mostly spina bifida, which occurs at very early spina cord development. However, results from chicken embryo model showed that there was an increase in histone H3K79Hcy expression during brain development in high-HTL-treated chickens while a decreased expression of H3K79Hcy was detected in samples from the normal group, suggesting that abnormal H3K79Hcy expression might lead to the occurrence of NTDs in chicken.”

#2 – Nice explanation of why modifications on H3 are the focus of this manuscript. Please include this brief explanation in the manuscript.

Response:

We are so glad that the reviewer is generally in agreement with our strategy to focus on H3 modification. We have added the following sentence in Line129-132 in the revised manuscript.

“The levels of homocysteinylation of the aforementioned histone H3 were further evaluated because H3K4me1 or H3K27ac play a key role during differentiation of human embryonic stem cells (hESCs) to neuroepithelium(Rada-Iglesias et al. 2011), while H3K9me2 is found to be participating in neuronal differentiation, and H3K9 methylation and H3K4 methylation are involved in the nervous system disease (Ryu et al. 2006; Shi 2007; Akbarian and Huang 2009).”

#4 – Again, please add the explanation to the manuscript.

Response:

Thanks for the reviewer’s suggestion. Since the main text is restricted to be no more than 5,000 words in total (Introduction, Results, Discussion), we made a minor modification in the revised manuscript which reads now as “Among over 300 genes and 14 epigenetic regulator genes connected with NTDs in the mouse, only *Cecr2*, *Smarca4* and *Dnmt3b* are founded in H3K79Hcy peak genes. Therefore, we focused our first set of experiments on *Smarca4*, *Cecr2*, and *Dnmt3b*.”

#5 – Thanks for adding the human samples details requested to the manuscript. I do feel that the anencephalic fetus should be excluded. Brain development is very different in an anencephalic fetus compared to normal development and the disorganized brain tissue is also open to the amniotic fluid (and thus at some point degrading). Any differences in brain samples could be related to this and therefore skew the results. The sample is still included in the manuscript, but not mentioned (it is in a reference). In fact, on line 458-459 the text specifies that the fetuses were diagnosed with spina bifida. If the anencephalic sample is included, that must be stated in the methods and results. Yes exencephaly and spina bifida are both NTDs, but there are also major differences.

Response:

Thanks for the reviewer’s suggestion. We have removed the anencephalic fetus in the revised manuscript.

#13 – When describing the chick model, it would be good to briefly mention that increases in Hcy or HTL do not lead to NTDs in mice. Since so much work has been done on mice and the KO mice for the 3 genes focused on are described in mice, I think it’s important to mention. It can be stated this is one of the reasons a chick model was used.

Response:

Thanks for the reviewer's suggestion. We have added this explanation in the revised manuscript as "A number of previous studies have demonstrated that increases in levels of Hcy or HTL do not lead to NTDs in mice."

From reviewer 2, response to question #1 – the authors have produced an interesting figure which could be included as supplementary data.

Response:

Thanks for the reviewer's suggestion. We have added this figure as Supplementary Table 2.

In several places (line 220 and Fig 5 Legend, line 899-900), the authors discuss treatment on chicks after they are "hatched". Do they mean after the egg is laid? Hatching is long after neurulation.

Response:

Thanks for the reviewer's question. We have modified the sentence as "After incubation for 28-30 hours, the embryos were injected with 0.5 μ l of 0.5 mM HTL."

Figure 1 – the added quantitation of the Western blot in 1c needs to be described in the legend. 1f – are all species samples brain tissue? Fetal brain tissue?

Response:

Thanks for the reviewer's question. The description of 1C added in the legend as "These test repeated for 3 times and the quantitation of the Western blot showed on right. In the BSA group, the relative K-Hcy levels were 1 ± 0.05 , 1.15 ± 0.10 ; 1.11 ± 0.78 . In the BSA-Hcy group, the relative K-Hcy levels were 10.88 ± 1.02 , 5.48 ± 0.34 ; 1.39 ± 0.21 ."

And in 1F, brain were used just in Gallus gallus, mouse and human fetal. We have modified the sentence to "Presence of H3 homocysteinylation in different species, including *D. melanogaster*, Zebra fish, Gallus gallus brain, mouse brain and human fetal brain, demonstrated using western blotting with rabbit polyclonal anti-Hcy and anti-H3 antibodies."

Line 217 "including similar NTC gene distribution on chromosomes as human embryos???" This was explained in the authors rebuttal, but it needs to be explained further in the manuscript.

Response:

We thank the reviewer for the positive feedback and we have added a sentence to explain it in Line 223-226 of the revised manuscript as "Chicken represents an appropriate animal model to analyze dynamics of neurulation, and has advantages over other models, including a short period of embryogenesis and low cost. The expression pattern of NTC genes is similar in chicken and human embryo, derived from a conservation in chromosomal localization of these genes."

Reference

- Akbarian, S., and H. S. Huang. 2009. 'Epigenetic regulation in human brain-focus on histone lysine methylation', *Biol Psychiatry*, 65: 198-203.
- Broch, O. J., and P. M. Ueland. 1984. 'Regional distribution of homocysteine in the mammalian brain', *Journal of Neurochemistry*, 43: 1755-7.
- Du, Y., T. Cai, T. Li, P. Xue, B. Zhou, X. He, P. Wei, P. Liu, F. Yang, and T. Wei. 2015. 'Lysine malonylation is elevated in type 2 diabetic mouse models and enriched in metabolic associated proteins', *Molecular & Cellular Proteomics*, 14: 227-36.
- Gurda, D., L. Marczak, and H. Jakubowski. 2014. 'Histones are targeted for N-homocysteinylation in human endothelial cells', *Acta Biochim Pol*, Pol 61 Suppl 1: 127.
- Marczak, L., M. Sikora, M. Stobiecki, and H. Jakubowski. 2011. 'Analysis of site-specific N-homocysteinylation of human serum albumin in vitro and in vivo using MALDI-ToF and LC-MS/MS mass spectrometry', *J Proteomics*, 74: 967-74.
- Rada-Iglesias, A., R. Bajpai, T. Swigut, S. A. Brugmann, R. A. Flynn, and J. Wysocka. 2011. 'A unique chromatin signature uncovers early developmental enhancers in humans', *Nature*, 470: 279-83.
- Ryu, H., J. Lee, S. W. Hagerty, B. Y. Soh, S. E. McAlpin, K. A. Cormier, K. M. Smith, and R. J. Ferrante. 2006. 'ESET/SETDB1 gene expression and histone H3 (K9) trimethylation in Huntington's disease', *Proc Natl Acad Sci U S A*, 103: 19176-81.
- Shi, Y. 2007. 'Histone lysine demethylases: emerging roles in development, physiology and disease', *Nature Reviews Genetics*, 8: 829-33.
- Xie, Z., D. Zhang, D. Chung, Z. Tang, H. Huang, L. Dai, S. Qi, J. Li, G. Colak, Y. Chen, C. Xia, C. Peng, H. Ruan, M. Kirkey, D. Wang, L. M. Jensen, O. K. Kwon, S. Lee, S. D. Pletcher, M. Tan, D. B. Lombard, K. P. White, H. Zhao, J. Li, R. G. Roeder, X. Yang, and Y. Zhao. 2016. 'Metabolic Regulation of Gene Expression by Histone Lysine beta-Hydroxybutyrylation', *Molecular Cell*, 62: 194-206.
- Xu, L., J. Chen, J. Gao, H. Yu, and P. Yang. 2015. 'Crosstalk of homocysteinylation, methylation and acetylation on histone H3', *Analyst*, 140: 3057-63.
- Zhang, Min, Lei Wang, Pei Pei, YiHua Bao, Jin Guo, Li Wang, ShaoYan Chang, XiaoLu Xie, HaiQin Cheng, Li Quan, and Ting Zhang. 2018. 'Development and clinical application of a LC-MS/MS method for simultaneous determination of one-carbon related amino acid metabolites in NTD tissues', *Analytical Methods*, 10: 1315-24.
- Zhao, S., W. Xu, W. Jiang, W. Yu, Y. Lin, T. Zhang, J. Yao, L. Zhou, Y. Zeng, H. Li, Y. Li, J. Shi, W. An, S. M. Hancock, F. He, L. Qin, J. Chin, P. Yang, X. Chen, Q. Lei, Y. Xiong, and K. L. Guan. 2010. 'Regulation of cellular metabolism by protein lysine acetylation', *Science*, 327: 1000-4.